# DISCRETE FEYNMAN-KAC CORRECTORS

## ABSTRACT

Discrete diffusion models have recently emerged as a promising alternative to the autoregressive approach for generating discrete sequences. Sample generation via gradual denoising or demasking processes allows them to capture hierarchical non-sequential interdependencies in the data. These custom processes, however, do not assume a flexible control over the distribution of generated samples. We propose DISCRETE FEYNMAN-KAC CORRECTORS, a framework that allows for controlling the generated distribution of discrete masked diffusion models at inference time. We derive Sequential Monte Carlo (SMC) algorithms that, given a trained discrete diffusion model, control the temperature of the sampled distribution (i.e. perform annealing), sample from the product of marginals of several diffusion processes (e.g. differently conditioned processes), and sample from the product of the marginal with an external reward function, producing likely samples from the target distribution that also have high reward. Notably, our framework does not require any training of additional models or fine-tuning of the original model. We illustrate the utility of our framework in several applications including: efficient sampling from the annealed Boltzmann distribution of the Ising model, improving the performance of language models for code generation and amortized learning, as well as reward-tilted protein sequence generation.

## 1 INTRODUCTION

The success of diffusion models in continuous domains, such as the generation of images (Rombach et al., 2022), videos (Wang et al., 2023; Blattmann et al., 2023), or 3D protein structures (Abramson et al., 2024; Watson et al., 2023), has motivated their application to discrete data spaces. Indeed, modeling discrete data such as text or biological sequences using diffusion processes is a promising direction since they do not rely on sequential token generation as with autoregressive models, which can impose arbitrary orderings on data (e.g., molecular structures and protein sequences (Lee et al., 2025; Alamdari et al., 2023)), or can suffer from exposure biases that limit long-horizon planning or reversal reasoning in natural language domains (Berglund et al., 2023; Nie et al., 2025).

Discrete diffusion is a general framework that defines a Continuous-Time Markov Chain (CTMC) process that progressively transforms data to a tractable distribution through a series of random transitions, and then learns to reverse this process and recover the original data distribution (Campbell et al., 2022; Lou et al., 2024; Sahoo et al., 2024; Shi et al., 2024). Furthermore, using external classifiers (Vignac et al., 2022; Nisonoff et al., 2024; Tang et al., 2025) or correction schemes (Nisonoff et al., 2024; Gruver et al., 2023) one can efficiently sample from various conditional distributions, e.g. conditioning on desired target properties of a protein (Gruver et al., 2023).

Most practical applications, however, require producing novel and task-specific generations rather than precise recreation of the training data. To produce novel generations, most generative models rely either purely on generalization abilities (Brown et al., 2020; Saharia et al., 2022) or on external reward functions in different forms (DeepSeek-AI, 2025; Rector-Brooks et al., 2024; Singhal et al., 2025). Furthermore, it has been shown that one can control the distribution of the produced samples by running task-specific Sequential Monte Carlo (SMC) methods at inference time (Skreta et al., 2024; 2025; He et al., 2025). In particular, Skreta et al. (2025) proposes the Feynman-Kac Correctors, which enable sampling from annealed densities ($p_t^{\text{anneal}}(x) \propto p_t(x)^\beta$) or a product of multiple densities ($p_t^{\text{prod}}(x) \propto \prod_{i=1}^M p_t^i(x)$) by simulating weighted stochastic differential equations (SDEs) with SMC resampling. This framework, however, is derived and presented only for the Fokker-Planck equation and does not directly apply to the discrete diffusion models, which are described by CTMC.

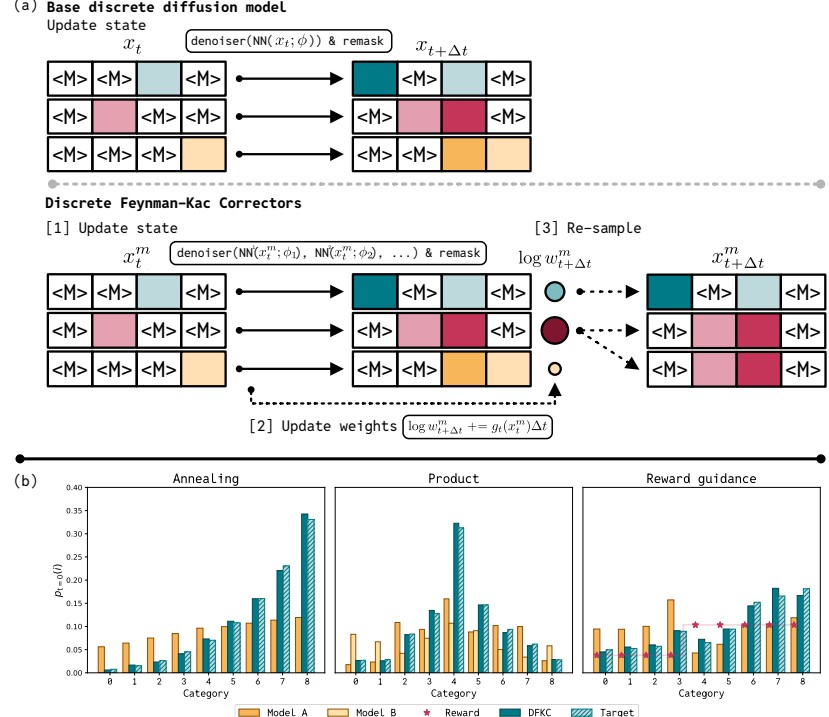

Figure 1: DISCRETE FEYNMAN-KAC CORRECTORS allow sampling from annealed distributions, product (or geometric average), and reward-tilted distributions. Panel (a) depicts the schematic of DFKC compared to the standard inference of masked discrete diffusion. Panel (b) demonstrates how DFKC, given trained discrete diffusion models and the reward function, samples from modified distributions at inference time.

We cover the existing literature gap by introducing DISCRETE FEYNMAN-KAC CORRECTORS (DFKC) — a principled framework enabling the control of discrete diffusion models at inference time (see Figure 1). In particular, given a trained discrete diffusion model with marginals $p_t(i)$ or several models with $p_t^1(i), p_t^2(i), \ldots$ (or the same model with different conditions $p_t(i \,|\, c_1), p_t(i \,|\, c_2), \ldots$), we modify the inference process to sample from the: (i) temperature annealed version of the marginals $p_t^{\text{anneal}}(i) \propto p_t(i)^\beta$, where $\beta$ is the inverse temperature (ii) product of corresponding marginals $p_t^{\text{prod}}(i) \propto p_t^1(i) p_t^2(i)$ (iii) geometric average of the marginals $p_t^{\text{avg}}(i) \propto p_t^1(i)^\gamma p_t^2(i)^{(1-\gamma)}$ (iv) reward-tilted marginals $p_t^{\text{reward}}(i) \propto p_t(i) \exp(\beta_t r(i))$, where $r(i)$ is the external reward function.

Our contribution is two-fold, we establish the theoretical framework that applies to general CTMC processes and we illustrate its utility with multiple applications on different domains. In particular, for each part of the framework, we choose the most promising and fitting applications: (i) we demonstrate that DFKC allows for efficient inference-time control of the temperature when sampling the configurations of the Ising model, which can be used as an efficient sampling algorithm (Akhound-Sadegh et al., 2025) (ii) we demonstrate that applying DFKC to a language model can be used to improve performance on programming tasks (with annealing) and allow scaling to larger prompts for amortized learning (using products) (iii) finally, we demonstrate how DFKC can be used to generate realistic protein sequences (Wang et al., 2024b) while optimizing external reward functions.

## 2 BACKGROUND

We consider continuous-time Markov chains (CTMC) or jump processes on discrete state spaces. Namely, every variable $x_t$ can take values in the range $0, \ldots, m$, and the time $t$ is in the interval $t \in [0, 1]$. All such processes are described by the Forward Kolmogorov Equation (FKE) (Kolmogoroff, 1931), which is why our main results are stated in terms of these equations.

For the discrete diffusion, we consider the specific case of masked diffusion processes and reserve a specific 'mask' state $m$ into the set of discrete states. We simulate the diffusion process by discretizing the corresponding FKE in time, and use the standard notation: $\text{Cat}(x \,|\, \pi)$ denotes the categorical distribution with probabilities $\pi$, $\delta_{ij}$ is the Kronecker symbol.

## 2.1 SIMULATING FORWARD KOLMOGOROV EQUATION (FKE)

The forward Kolmogorov equation for continuous-time Markov chains describes the evolution of the transition probability as follows

$$\frac{\partial p(x_s = j \mid x_t = i)}{\partial s} = \sum_k A_s(k, j) p(x_s = k \mid x_t = i) \,, \ A_s(k, j) := \left. \frac{\partial p(x_t = j \mid x_s = k)}{\partial t} \right|_{t=s}.$$

In practice, FKE can be used to parameterize the time-evolution of the marginals by specifying the rate matrix $A_t(i, j)$ and the initial boundary condition $p_{t=0}(i) := p(x_0 = i)$. In this case, the change of the marginals is defined as follows

$$\frac{\partial p_t(i)}{\partial t} = \sum_j A_t(j, i) p_t(j) \,, \ \sum_j A_t(i, j) = 0 \,, \ A_t(i, i) \le 0 \,, A_t(i, j) \ge 0 \,, \ \forall i \ne j \,, \quad (1)$$

where we introduce constraints on the family of the possible matrices $A_t(i, j)$ according to the definition of the rate matrix.

Fortunately, this constraints can be easily satisfied by parameterizing only the off-diagonal terms of the matrix $A_t(i, j)$ and defining the diagonal term $A_t(i, i)$ as the negative sum over the off-diagonal.

$$\frac{\partial p_t(i)}{\partial t} = \sum_{j \ne i} (A_t(j, i) p_t(j) - A_t(i, j) p_t(i)) \,. \quad (2)$$

To draw samples from $p_t(i)$ one can draw samples from $p_0(i)$ and simulate FKE by discretizing it in time. Namely, at every iteration, one samples from the following conditional probability

$$p(x_{t+dt} = j \mid x_t = i) = \delta_{ij} + A_t(i, j) dt + o(dt) \,, \text{ i.e. } x_{t+dt} \sim \texttt{Cat}(x_{t+dt} = j \mid \delta_{ij} + A_t(i, j) dt) \,. \quad (3)$$

In this work, we are interested in FKEs of the particular form

$$\frac{\partial p_t(i)}{\partial t} = \sum_{j \ne i} (A_t(j, i) p_t(j) - A_t(i, j) p_t(i)) + p_t(i) \big( g_t(i) - \mathbb{E}_{p_t(i)} g_t(i) \big) \,, \quad (4)$$

where the first term corresponds to the standard FKE as in Equation (2) and the second term corresponds to re-weighting of the samples according to some function $g_t(i)$. In general, the second term does not extend the family of jump processes described by the standard FKE because it can be incorporated into the rate matrix (see Appendix B.1). However, importantly, this term allows using the Feynman-Kac formula as stated in the following theorem (see the derivation in Appendix B.2).

**Theorem 2.1.** [Feynman-Kac Formula] *For the forward Kolmogorov equation from Equation* (4) *describing the time-evolution of the marginals $p_t(i)$ with the rate matrix $A_t(i, j)$ and weights $g_t(i)$, $\bar{g}_t(i) = g_t(i) - \sum_k p_t(k) g_t(k)$*

$$\mathbb{E}_{p_T(x)} \phi(x) = \lim_{dt \to 0} \sum_{x_T} \dots \sum_{x_0} \phi(x_T) p(x_T \mid x_{T-dt}) \dots p(x_{dt} \mid x_0) \exp \left( \sum_{t=0}^{T} dt \bar{g}_t(x_t) \right) p_0(x_0)$$

$$= \mathbb{E}_{X_{0:T}} \exp \left( \int_0^T dt \ \bar{g}_t(X_t) \right) \phi(X_T) \propto \mathbb{E}_{X_{0:T}} \exp \left( \int_0^T dt \ g_t(X_t) \right) \phi(X_T) \,, \quad (5)$$

*where the expectation on the right hand side is taken w.r.t. trajectories $X_{0:T}$ defined as the limit of the transitions from Equation* (3).

In particular, to simulate Equation (4), one can extend the states $x_t$ with the weights $w_t$ and jointly simulating the following equations

$$\text{for } x_t = i \,, \ x_{t+dt} \sim \texttt{Cat}(x_{t+dt} = j \mid \delta_{ij} + A_t(i, j) dt) \,, \ \log w_{t+dt} = \log w_t + g_t(i) dt \,. \quad (6)$$

Finally, the weighted samples $(x_T^k, w_T^k)$ can be used for the Self-Normalized Importance Sampling (SNIS) estimator or the corresponding empirical measure

$$\mathbb{E}_{p_T(i)}\phi(i) \approx \sum_k \frac{w_T^k}{\sum_j w_T^l}\phi(x_T^k), \quad p_T(i) \approx \sum_k \frac{w_T^k}{\sum_l w_T^l}\delta_{ix_T^k}. \tag{7}$$

## 2.2 DISCRETE MASKED DIFFUSION

Analogously to continuous-space diffusion models (Song et al., 2021), the discrete diffusion models operate by mapping the data distribution $p_0(i)$ to a simple marginal $p_1(i)$ and then simulating the reverse process. In particular, masked diffusion models define a conditional probability $p(x_s = j \mid x_t = i)$ as a probability of switching from any state to the $m$-th state, which denotes the utility 'mask' state. These conditional probabilities can be described using the following formula (see the derivation in Appendix B.3), which yields the corresponding rate matrix.

$$p(x_s = j \mid x_t = i) = \left(1 - \frac{\alpha_s}{\alpha_t}\right)\delta_{mj} + \frac{\alpha_s}{\alpha_t}\delta_{ij}, \quad A_t(i, j) = \frac{1}{\alpha_t}\frac{\partial\alpha_t}{\partial t}(\delta_{ij} - \delta_{mj}) \tag{8}$$

In general, the reverse-time process with the marginals $q_\tau(i) := p_{1-\tau}(i)$ is also described by FKE

$$\frac{\partial q_\tau(i)}{\partial \tau} = \sum_{j \neq i}(B_\tau(j, i)q_\tau(i) - B_\tau(i, j)q_\tau(i)), \quad B_\tau(i, j) = A_{1-\tau}(j, i)\frac{p_{1-\tau}(j)}{p_{1-\tau}(m)}, \tag{9}$$

where $A_t(i, j)$ and $B_\tau(i, j)$ are the rate matrices of the forward-time and reverse-time processes correspondingly (see Appendix B.4). Note that here and throughout the paper we define only the off-diagonal terms of the matrices and the diagonal is automatically defined as $B_\tau(i, i) = -\sum_{j \neq i} B_\tau(i, j)$.

Finally, one can sample from the data distribution $p_{t=0}(i)$ by first generating samples from $p_{t=1}(i)$ and then simulating the reverse-time FKE from Equation (9). For the masked diffusion process from Equation (8) the off-diagonal elements of the rate matrix are

$$B_\tau(i, j) = -\delta_{mi}\frac{1}{\alpha_t}\frac{\partial\alpha_t}{\partial t}\frac{p_t(j)}{p_t(m)} = -\delta_{mi}\frac{1}{\alpha_t}\frac{\partial\alpha_t}{\partial t}\left(\delta_{mj} + \frac{\alpha_t}{1 - \alpha_t}p(x_0 = j \mid x_t = m)\right), \tag{10}$$

where the last equality (shown in Shi et al. (2024)) comes from the relation between the ratio of probabilities $p_t(j)/p_t(m)$ and the conditional de-masking probability $p(x_0 = j \mid x_t = m)$ (see details in Appendix B.5). In practice, one can parameterize either 'score' $s_t(m, j; \theta) = p_t(j)/p_t(m)$ (as suggested in Lou et al. (2024); Benton et al. (2024)) or the de-masking probability $p(x_0 = j \mid x_t = m) = (1 - \delta_{mj})\texttt{softmax}(\texttt{NN}(x_t; \theta))_j$ (as suggested in (Shi et al., 2024)). For our purposes, these parameterization are equivalent. Furthermore, both these parameterizations can be learned by maximizing the same Evidence Lower Bound (ELBO) objective.

Finally, all the derivations seamlessly transfer to any number of dimensions (see Appendix B.6). In particular, one can define the masking process independently over the dimensions, and obtain the following off-diagonal elements of the reverse-time rate matrix

$$B_t(i_1 \ldots i_d, j_1 \ldots j_d) = -\frac{1}{\alpha_t}\frac{\partial\alpha_t}{\partial t}\frac{p_t(j_1 \ldots j_d)}{p_t(i_1 \ldots i_d)}\sum_{k=1}^d \prod_{l \neq k}\delta_{j_l i_l}\delta_{mi_k}, \quad [i_1 \ldots i_d] \neq [j_1 \ldots j_d], \tag{11}$$

which are not zero only when all the coordinates except one match. Thus, one can parameterize the reverse-time process by predicting $(m - 1)d$ values, where $d$ is the number of dimensions (or sequence length) and $(m - 1)$ is the vocabulary size for each discrete variable.

## 3 DISCRETE FEYNMAN-KAC CORRECTORS

In this section, we introduce DISCRETE FEYNMAN-KAC CORRECTORS— a framework that allows for inference-time control of discrete diffusion models. Our derivations proceed in the same fashion for all the cases. First, we consider general CTMC processes with given rate matrices and initial conditions, which induce corresponding marginals. Applying different transformations to these marginals (annealing, product, geometric averaging, reward-tilting), we define new CTMC processes and derive corresponding rate matrices. These derivations state our main results in the most general

form. Further, we proceed by applying these derivations to the masked diffusion processes and demonstrate that the transformed processes can be efficiently simulated without any additional training or finetuning. For each case, as we demonstrate, one requires only the ratio of marginal densities, or, equivalently, the denoising conditional probability, which are used for parameterizing the reverse-time process as shown in Equation (10).

## 3.1 TEMPERATURE ANNEALING[1]

First, we present the general result that holds for the forward Kolmogorov equation with arbitrary rate matrix $A_t(i, j)$. Since we do not assume any structure of the matrix, it is easier to reason in terms of Equation (2), i.e. using only the off-diagonal entries assuming that the diagonal elements are chosen correspondingly to define the correct rate matrix. The annealed FKE is as follows.

**Theorem 3.1.** [Temperature Annealing] *Consider the forward Kolmogorov equation from Equation (2) describing the time-evolution of the marginals $p_t(i)$ with the rate matrix $A_t(i, j)$. For the temperature annealed marginals $q_t(i) \propto p_t(i)^\beta$, the following equation holds*

$$\frac{\partial q_t(i)}{\partial t} = \sum_{j \neq i} \left( A_t^{\mathrm{anneal}}(j, i) q_t(j) - A_t^{\mathrm{anneal}}(i, j) q_t(i) \right) + q_t(i) \left( g_t(i) - \mathbb{E}_{q_t(j)} g_t(j) \right), \quad (12)$$

*where* $A_t^{\mathrm{anneal}}(i, j) \coloneqq \beta A_t(i, j) \dfrac{p_t^{1-\beta}(i)}{p_t^{1-\beta}(j)}$, $g_t(i) \coloneqq \sum_{j \neq i} \left( A_t^{\mathrm{anneal}}(i, j) - \beta A_t(i, j) \right)$. (13)

Thus, the annealed FKE relies on the rate matrix $A_t(i, j)$ of the original process and the ratio of marginal probabilities $p_t(i)/p_t(j)$, which are readily available for a trained model of the masked diffusion process. The following corollary presents the rate matrix and the weighting function for the reverse-time masked diffusion process.

**Corollary 3.2.** [Annealed Masked Diffusion] *For the rate matrix of the reverse-time masked diffusion from Equation (10), Theorem 3.1 yields the following off-diagonal elements of the rate matrix and the corresponding weight function*

$$B_\tau^{\mathrm{anneal}}(i, j) = -\delta_{mi} \frac{\beta}{\alpha_t} \frac{\partial \alpha_t}{\partial t} \frac{p_t^\beta(j)}{p_t^\beta(m)}, \quad g_\tau(i) = \delta_{mi} \frac{\beta}{\alpha_t} \frac{\partial \alpha_t}{\partial t} \sum_j \left( \frac{p_t(j)}{p_t(m)} - \frac{p_t^\beta(j)}{p_t^\beta(m)} \right). \quad (14)$$

This corollary demonstrates that both the new rate matrix and the weights can be efficiently evaluated using the ratio of the marginals, which is used in practice to parameterize the reverse process (see Equation (10)). In more detail, one can obtain the new rate matrix by simply scaling it by $\beta$ and raising the probability ratio to the power $\beta$

$$\frac{p_t^\beta(j)}{p_t^\beta(m)} = \delta_{mj} + \frac{\alpha_t^\beta}{(1 - \alpha_t)^\beta} \exp(\beta \log p(x_0 = j \mid x_t = m)), \quad (15)$$

which corresponds to multiplying the logits of the denoising model by $\beta$ besides adjusting the schedule dependent coefficients. Finally, the weighting term can be easily obtained by the summation of the probability ratios $p_t(j)/p_t(m)$ over $j$, which corresponds to the summation over the different coordinates of the network output and does not require additional function evaluations.

## 3.2 PRODUCT AND GEOMETRIC AVERAGING[2]

Sampling from the product of marginals can be interpreted as generating samples that are likely according to several models at the same time. Intuitively, all the models must "unanimously agree" on the sample being likely since zero probability of one of the models renders the entire product to be zero (Hinton, 1999). In what follows, we formalize this collaborative generation process as the process with marginals proportional to the product of marginals of different CTMC processes and state it in the general case with arbitrary rate matrices. For simplicity, here, we present the results

---

[1]See Appendix C.1 for the proofs
[2]See Appendix C.2 for the proofs

for the product of two marginals and postpone the general formulation for geometric average of any number of the marginals to Theorem C.3 and Theorem C.4 in Appendix C.3.

> **Theorem 3.3.** [Product of FKEs] *Consider two forward Kolmogorov equations (from Equation* (2)*) with different rate matrices $A_t^1(i,j)$ and $A_t^2(i,j)$ describing the evolution of marginals $p_t^1(i)$ and $p_t^2(i)$. For the product of marginals $q_t(i) \propto p_t^1(i)p_t^2(i)$, the following equation holds*
>
> $$\frac{\partial q_t(i)}{\partial t} = \sum_{j \neq i} \left( A_t^{\mathrm{prod}}(j,i)q_t(j) - A_t^{\mathrm{prod}}(i,j)q_t(i) \right) + q_t(i)\left( g_t(i) - \mathbb{E}_{j \sim q_t(j)} g_t(j) \right), \qquad (16)$$
>
> $$A_t^{\mathrm{prod}}(i,j) := A_t^1(i,j)\frac{p_t^2(j)}{p_t^2(i)} + A_t^2(i,j)\frac{p_t^1(j)}{p_t^1(i)} \,,\ g_t(i) := \sum_{j \neq i} \left( A_t^{\mathrm{prod}}(i,j) - A_t^1(i,j) - A_t^2(i,j) \right).$$

Importantly, the new rate matrix and the weighting terms are defined in terms of both rate matrices $A_t^1(i,j)$ and $A_t^2(i,j)$ and the ratios of probabilities $p_t^1(i)/p_t^1(j)$ and $p_t^2(i)/p_t^2(j)$. All these quantities are readily available in the masked diffusion models. To be precise, we present the corresponding reverse-time rate matrix and the weighting term in the following corollary.

> **Corollary 3.4.** [Product of Masked Diffusions] *For the rate matrix of the reverse-time masked diffusion from Equation* (10)*, Theorem 3.3 yields*
>
> $$B_\tau^{\mathrm{prod}}(i,j) = -2\delta_{mi}\frac{1}{\alpha_t}\frac{\partial \alpha_t}{\partial t}\frac{p_t^1(j)}{p_t^1(m)}\frac{p_t^2(j)}{p_t^2(m)} \,, g_\tau(i) = \frac{\delta_{mi}}{\alpha_t}\frac{\partial \alpha_t}{\partial t}\sum_j \frac{p_t^1(j)}{p_t^1(m)} + \frac{p_t^2(j)}{p_t^2(m)} - 2\frac{p_t^1(j)}{p_t^1(m)}\frac{p_t^2(j)}{p_t^2(m)}$$

According to these formulas, both the rate matrix and the weights can be efficiently evaluated with a single forward pass through each network.

## 3.3 Reward-tilted Marginals[3]

Generative modeling allows optimizing the external reward functions $r(i)$ while staying within the data distribution $p_{t=0}(i)$ to avoid over-optimization and collapsing to degenerate solutions. Usually it is formalized as sampling from the reward-tilted distribution $p_{t=0}(i)\exp(r(i))$, which we discuss in this section. The following result modifies any CTMC process to sample from the reward-tilted distribution. Note that we derive formulas for the off-diagonal elements of the rate matrix.

> **Theorem 3.5.** [Reward-tilted FKE] *Consider the forward Kolmogorov equation from Equation* (2) *describing the time evolution of the marginals $p_t(i)$ with the rate matrix $A_t(i,j)$. For the reward-tilted marginals $q_t(i) \propto p_t(i)\exp(\beta_t r(i))$, the following equation holds*
>
> $$\frac{\partial q_t(i)}{\partial t} = \sum_{j \neq i} \left( A_t^{\mathrm{reward}}(j,i)q_t(j) - A_t^{\mathrm{reward}}(i,j)q_t(i) \right) + q_t(i)\left( g_t(i) - \mathbb{E}_{q_t(j)} g_t(j) \right), \quad (17)$$
>
> $$A_t^{\mathrm{reward}}(i,j) := A_t(i,j)\frac{\exp(\beta_t r(j))}{\exp(\beta_t r(i))} \,,\ g_t(i) := \sum_{j \neq i} \left( A_t^{\mathrm{reward}}(i,j) - A_t(i,j) \right) + \frac{\partial \beta_t}{\partial t}r(i). \quad (18)$$

Note that the obtained formulas depend only on the reward function and the rate matrix of the original process. Applying this result to the masked diffusion we obtain the following corollary.

> **Corollary 3.6.** [Reward-tilted Masked Diffusion] *For the rate matrix of the reverse-time masked diffusion from Equation* (10)*, Theorem 3.5 yields*
>
> $$B_\tau^{\mathrm{reward}}(i,j) = -\delta_{mi}\frac{1}{\alpha_t}\frac{\partial \alpha_t}{\partial t}\frac{p_t(j)}{p_t(m)}\frac{\exp(\beta_t r(j))}{\exp(\beta_t r(m))} \,, \qquad (19)$$
>
> $$g_\tau(i) = \frac{1}{\alpha_t}\frac{\partial \alpha_t}{\partial t}\delta_{mi}\sum_j \left( \frac{p_t(j)}{p_t(m)} - \frac{p_t(j)}{p_t(m)}\frac{\exp(\beta_t r(j))}{\exp(\beta_t r(m))} \right) + \frac{\partial \beta_t}{\partial t}r(i). \qquad (20)$$

---

[3]See Appendix C.4 for the proofs

---

**Algorithm 1:** Generation using DISCRETE FEYNMAN-KAC CORRECTORS

---

**Input:** corresponding rate matrix $B_\tau(i,j)$ and weight function $g_\tau(i)$, number of samples $K$

1   $x^k_{\tau=0} \sim p_{t=1}(i)$;                                         `/* initialize with noise */`

2   $w^k_{\tau=0} = 1/K$;                                             `/* uniform weights */`

3   **for** $\tau = 0, \dots, 1$ **do**

4      $x^k_{\tau+d\tau} \sim \mathtt{Cat}(x^k_{\tau+d\tau} = j \mid \delta_{ij} + B_\tau(i,j)d\tau)$, for $x^k_\tau = i$ ;     `/* update state */`

5      $\log w^k_{\tau+d\tau} = \log w^k_\tau + g_\tau(i)d\tau$ ;                   `/* update weights */`

6      **if** *resample* **then**

7          $w^k_{\tau+d\tau} = w^k_{\tau+d\tau}/\left(\sum_l w^l_{\tau+d\tau}\right)$ ;          `/* re-normalize weights */`

8          $x^k_{\tau+d\tau} = x^\ell_{\tau+d\tau}$, $\ell \sim \mathtt{Cat}(l \mid w_{\tau+d\tau})$ ;         `/* re-sample indices */`

9          $w^k_{\tau+d\tau} = 1/K$;                                `/* re-initialize weights */`

**Output:** weighted set of samples $\{(x^k_{\tau=1}, w^k_{\tau=1})\}^K_{k=1}$

---

Note that evaluating $B^{\mathrm{reward}}_\tau(i,j)$ requires computing the reward function at all the states $j$ we can transition to from mask $m$. Furthermore, computing $g_t(i)$ requires the summation of the reward over all such states $j$, which, depending on the application, might be computationally expensive. To avoid these extra computations one could potentially use alternative functions evaluating the difference in the rewards on the transitions from $m$ to $j$, i.e. $r(j) - r(m)$. However, we leave this as a future work.

## 4   EXPERIMENTS

In this section, we demonstrate the utility of the proposed DISCRETE FEYNMAN-KAC CORRECTORS on several applications using modern discrete diffusion models. Each experiment is aimed at illustrating one of the introduced processes: annealing, geometric averaging, reward-tilting.

Despite different domains and processes, the generation process always follows the same procedure described in Alg. 1. Namely, for the corresponding rate matrix $B_\tau(i,j)$ and weight function $g_\tau(i)$ (see Section 3 for their definitions), the inference procedure generates a batch of samples $x^k_\tau$ together with their weights $w^k_\tau$. In practice, we always perform resampling in between the update steps using SNIS. Thus, DFKC not only changes the generation of individual samples by changing the rate matrix $B_\tau(i,j)$ but also introduces "interactions" between samples through re-weighting and re-sampling. In Appendix A.1, we provide the explicit state and weight update rules for sampling from different target densities and show how these discrete updates correspond to the continuous formulation of Skreta et al. (2025).

### 4.1   ANNEALING THE ISING MODEL

We apply Theorem 3.1 for annealing the Boltzmann distribution of the Ising model configurations. Namely, the probability distribution of states $\sigma$ is given as

$$p_\beta(\sigma) = \frac{1}{Z_\beta} e^{-\beta H(\sigma)}, \; Z_\beta = \sum_\sigma e^{-\beta H(\sigma)}, \; \text{where } H(\sigma) = -\sum_{i,j} J_{ij}\sigma_i\sigma_j - \sum_i h_i\sigma_i. \quad (21)$$

We generate a training dataset at a fixed $\beta$ by running the Swendsen-Wang algorithm (Swendsen & Wang, 1987) and train a discrete masked-diffusion model. We set $J_{ij} = 1$ and $h_i = 0$ on a 16×16 lattice with open boundary conditions. The diffusion model is parameterized using the UNet architecture. We assess method performance by comparing the distributions of key observables, specifically energy and magnetization. To examine the fidelity of local structures, we compute spin–spin correlations as a function of distance, excluding boundary spins and evaluating correlations along lattice rows. Finally, we evaluate the mean squared error (MSE) between the generated correlation profiles and the ground-truth.

We train the diffusion model at $\beta = 0.25$ and demonstrate that DFKC allows for the efficient control of temperature at inference time in the range $\beta \in [0.25, 0.6]$, with critical point $\beta_{\mathrm{crit}} \approx 0.4407$. As a baseline, we consider a guidance method, which ignores the weights of the generated samples. In addition, a comparison with LEAPS (Holderrieth et al., 2025) is provided. Note that the released LEAPS model is trained on the annealing path $\pi_t(\sigma) \propto \exp\left(-t\,\beta_{\mathrm{crit}} H(\sigma)\right)$ for $t \in [0, 1]$, so we do

Table 1: Sampling task for Ising model with performance measured by mean ±standard deviation over 3 seeds. The starting temperature for DFKC is shown in brackets. The DDM samples are generated with a discrete diffusion model trained at those corresponding target temperatures.

| Target $\beta$ | Method | Energy-$\mathcal{W}_2(\downarrow)$ | Magnetization-$\mathcal{W}_2(\downarrow)$ | Correlation-MSE ($\downarrow$) |
|---|---|---|---|---|
| 0.4 | DFKC(0.3) | $14.24 \pm 3.11$ | $0.256 \pm 0.052$ | $0.041 \pm 0.013$ |
| | DDM | $69.38 \pm 4.25$ | $0.889 \pm 0.063$ | $0.172 \pm 0.021$ |
| 0.3 | DFKC(0.2) | $33.38 \pm 0.46$ | $0.031 \pm 0.011$ | $0.023 \pm 0.007$ |
| | DDM | $35.14 \pm 0.63$ | $0.046 \pm 0.012$ | $0.014 \pm 0.009$ |

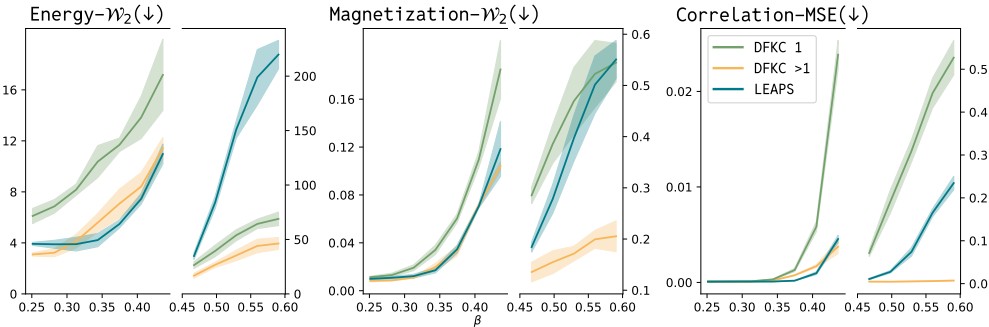

Figure 2: 2-Wasserstein metric for energy and magnetization distributions and MSE for spin-spin correlation. All metrics are computed between samples from DFKC variants or LEAPS and samples from Swendsen-Wang algorithm. The $\beta$ used for training DFKC is 0.25. Note that the plots include a break at $\beta_{\text{crit}}$, and the $y$-axes use different scales on either side of $\beta_{\text{crit}}$ to make differences in magnitude visible.

not expect it to work reliably for $\beta > \beta_{\text{crit}}$, and instead expect it to sample only within the range of temperatures it was trained on. Notably, our temperature annealing method, being training-free, allows for sampling beyond the critical temperature. The results in Figure 2 validate this perspective.

In Table 1, we demonstrate that collecting the data at a high temperature and annealing the trained model to the low temperature is more efficient than collecting data and training the model directly at a low temperature. In particular, we fix the number of energy evaluations for the dataset collection and can either allocate this budget for training the discrete diffusion model (DDM) directly on the target temperature, or for training it a higher temperature and then using DFKC to reduce the temperature to the target. To conduct this comparison, we used 10,000 samples following a long burn-in period of Glauber dynamics, which requires lengthy chains to reduce correlations. Additional details of the experiments are included in Appendix D.5.

## 4.2 Products and Annealing for Language Modelling

We evaluate the ability of DFKC to improve performance on text generation tasks by evaluating the annealing formula (from Theorem 3.1) for code generation, and the product formula from Theorem 3.3 for amortized learning. For both tasks, we use the pretrained LLaDA-8B-Instruct as our (masked) diffusion model (Nie et al., 2025).

**Code Generation.** Recent work argues that annealing an existing model to sample higher likelihood points can result in better performance on various mathematical or coding tasks (Huang et al., 2024). Inspired by this line of work, we investigate the applicability of our annealing method as an inference time strategy for improving accuracy on coding tasks. We anneal the model with parameter $\beta$ and evaluate its accuracy in solving a diverse set of programming problems in the HumanEval and MBPP datasets (Chen et al., 2021; Austin et al., 2021). We compare with sampling the most likely token at each step ("Argmax" sampling), as well as sampling with inverse-temperature $\beta$ ("Naive Annealing"). The results are reported in Figure 4, which demonstrate that DFKC obtains a higher accuracy than other sampling methods. Additional details are included in Appendix D.2.

**Amortized Learning.** Given a dataset of examples $\mathcal{X} = \{(x_i, y_i)\}_{i=1}^N$, and a parametric model $f_\theta(x)$, we wish to use the language model to infer parameters $\theta$ which fit the data. This requires sampling from the posterior distribution over parameters $p(\theta|\mathcal{X})$. However, unlike more classical statistical methods, we wish to perform this computation solely through the text interface of the language

Table 2: Evaluation for reward-guided protein sequence generation (over 5 seeds).

| | Reward | Diversity | | Structural confidence | | | Novelty |
|---|---|---|---|---|---|---|---|
| | $\log r(x)$ ($\uparrow$) | Seq. diversity ($\uparrow$) | Max. cluster ($\uparrow$) | pLDDT ($\uparrow$) | pTM ($\uparrow$) | Frac. (pLDDT & pTM) $> 0.7$ ($\uparrow$) | Frac. BLASTp hits ($\downarrow$) |
| *Task: unconditional generation* | | | | | | | |
| Base [unguided] | $-0.4520 \pm 0.0676$ | $\mathbf{0.7729 \pm 0.0570}$ | $\mathbf{0.3333}$ | $0.5827 \pm 0.1539$ | $0.2535 \pm 0.1403$ | $0.00$ | $\mathbf{0.0267}$ |
| DG-Exact Nisonoff et al. (2024) | $-0.3225 \pm 0.0751$ | $0.7527 \pm 0.0765$ | $\mathbf{0.3333}$ | $0.5922 \pm 0.1540$ | $0.2653 \pm 0.1399$ | $0.00$ | $0.0733$ |
| FK Steering Singhal et al. (2025) | $-0.3553 \pm 0.0929$ | $0.6978 \pm 0.2167$ | $0.1067$ | $0.5765 \pm 0.1686$ | $0.2098 \pm 0.1168$ | $0.00$ | $0.0333$ |
| DFKC [ours] | $\mathbf{-0.2259 \pm 0.0415}$ | $0.6144 \pm 0.2954$ | $0.0533$ | $\mathbf{0.7148 \pm 0.1416}$ | $\mathbf{0.5006 \pm 0.2120}$ | $\mathbf{0.19}$ | $0.4133$ |
| *Task: thermostability* | | | | | | | |
| Base [unguided] | $-0.6590 \pm 0.1026$ | $0.7723 \pm 0.0582$ | $\mathbf{0.3571}$ | $0.5941 \pm 0.1525$ | $0.2609 \pm 0.1452$ | $0.00$ | $\mathbf{0.0286}$ |
| DG-Exact Nisonoff et al. (2024) | $-0.6131 \pm 0.1211$ | $\mathbf{0.7860 \pm 0.0590}$ | $\mathbf{0.3571}$ | $\mathbf{0.6088 \pm 0.1500}$ | $\mathbf{0.2695 \pm 0.1550}$ | $\mathbf{0.01}$ | $0.0857$ |
| FK Steering Singhal et al. (2025) | $-0.5841 \pm 0.1360$ | $0.7513 \pm 0.1723$ | $0.2733$ | $0.5704 \pm 0.1534$ | $0.2246 \pm 0.1087$ | $0.00$ | $0.0400$ |
| DFKC [ours] | $\mathbf{-0.5316 \pm 0.1082}$ | $0.7618 \pm 0.1302$ | $0.3200$ | $0.5875 \pm 0.1517$ | $0.2468 \pm 0.1387$ | $\mathbf{0.01}$ | $0.0533$ |

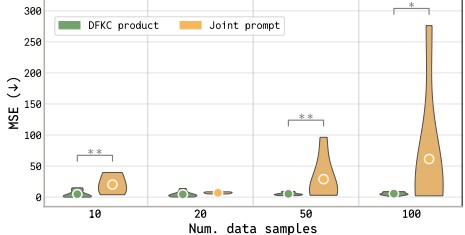

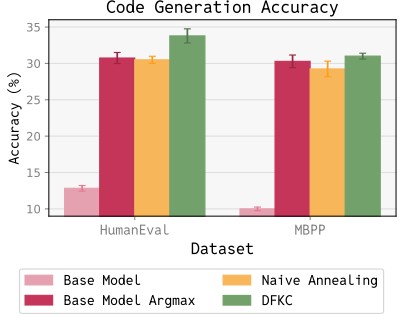

Figure 3: Amortized learning task: Mean squared error (MSE) between predicted and true parameters reported for DFKC (1 and 5 samples), and joint prompting, across different dataset sizes. $\star\star$ indicates $p \leq 0.02$, $\star$ indicates $p \leq 0.05$ (one-sided Student's $t$-test).

Figure 4: Accuracy on coding tasks, with standard error reported over 5 seeds.

model, similar to the setting of (Requeima et al., 2024; Mittal et al., 2025). Namely we set $\mathcal{X}$ as our prompt, and ask the model to sample parameters $\theta$. We partition the dataset into $K$ equal subsets $\mathcal{X} = \bigcup_{k=1}^{K} \mathcal{X}_k$, and note that for a uniform prior, the posterior factors as $p(\theta|\mathcal{X}) \propto \prod_{k=1}^{K} p(\theta|\mathcal{X}_k)$. This justifies applying our method, with each factor in the product conditioned on a different subset of the data $C_k = \mathcal{X}_k$. We evaluate this task on a synthetic dataset generated using a noisy linear predictor $f_\theta(x) = \theta_1 x + \theta_0 + \epsilon$ $\epsilon \sim \mathcal{N}(0, 0.1^2)$. We use $K = 5$ subsets, and report our results for the mean-squared error (to the true parameters) across larger datasets $\mathcal{X}$ in Figure 3. From our results, we can see that as the length and complexity of the prompt increases, the joint prompt degrades in performance, compared to the more stable performance of the DFKC product. We also see that using more samples in our method improves performance slightly over 1 sample. This is also validated by an ablation over the number of SMC samples in Figure A1. Additional details and results are included in Appendix D.1. In Appendix D.3, we present a related experiment which applies the product formulation of our method for generating stories adhering to a set of constraints.

## 4.3 GUIDING PROTEIN SEQUENCE GENERATION WITH EXTERNAL REWARDS

Finally, we investigate the utility of DFKC in the setting of unconditional *de novo* protein sequence generation. Protein language models (PLMs) have emerged as powerful tools for modeling the complex relationships between protein sequence, structure, and function (Lin et al., 2023; Madani et al., 2023), but controlling their outputs remains a significant challenge. To encourage generation of sequences that resemble natural proteins, we guide sampling with rewards that measure sequence plausibility. We consider two reward settings: (i) the likelihood under a PLM, and (ii) the predicted thermostability of the sequence. The likelihood reward is motivated by the fact that PLMs capture evolutionary constraints and assign higher probability to "natural-like" sequences, a property that has been successfully leveraged to steer generation toward functional and biologically viable proteins (Ertelt et al., 2024; Emami et al., 2023; Notin et al., 2023). The thermostability reward reflects that high stability is a desirable property of natural proteins, correlating with improved folding, robustness, and mutational tolerance. For likelihood we use the masked language model ESM2-650M (Lin et al., 2023), and for thermostability we use a fine-tuned version of DPLM-650M (Wang et al., 2024a).

We generate sequences using DPLM-650M, a discrete diffusion model that produces protein sequences by progressively unmasking amino acid tokens (Wang et al., 2024a). To guide generation, we set the reward appropriately and apply Theorem 3.5. Table 2 presents the rewards and additional metrics for sequences sampled using our method, for both tasks. The table also compares our method with other guidance-based techniques: FK Steering (using the base model as a proposal) (Singhal et al., 2025), and DG-Exact, the exact guidance approach of Nisonoff et al. (2024). In the guided setting,

we note that the single-sample variant of DFKC is equivalent to DG-Exact, and observe that using multiple samples yields notable improvements in mean reward compared to both unguided DPLM sampling and guidance without resampling. These results highlight the effectiveness of our resampling procedure in enhancing desired properties of generated sequences. For likelihood based sampling, our method generates more designable sequences, at the cost of diversity, while for thermostability it maintains similar performance to the base model across other metrics. DG-Exact requires $O(V)$ reward evaluations (with vocabulary size $V$) for each inference step, while FK Steering performs inference with $M$ samples in parallel. Our method uses both strategies, with run-time similar to DG-Exact (due to parallel computation over $M$). The combination allows for substantial reward improvement, and makes the method useful for tasks where additional inference compute can be spent to obtain higher quality samples. Additional experimental details and results are included in Appendix D.4.

## 5 RELATED WORK

**Reward Fine-tuning.** These methods often assume an external reward function $r(x)$ and adjust the pretrained model's parameters using reinforcement learning algorithms, with the goal of sampling from the product $r(x)q_t(x)$. Several of these works are applicable to discrete diffusion models (Venkatraman et al., 2024; Rector-Brooks et al., 2024; Wang et al., 2025). Our method leaves the pretrained model fixed, and therefore doesn't require a costly fine-tuning stage. We note that our method is compatible with a model obtained from reward fine-tuning, or from any training approach resulting in a parameterized rate matrix (Le-Tuyet-Nhi et al., 2025).

**Inference Time Alignment.** Several methods perform additional computation at inference time to sample from a target product distribution (the product being taken with either an external model $r(x)$, or a classifier extracted from the model's distribution, $q_t(y|x)$ as in classifier-free guidance (Ho & Salimans, 2022)). These methods often involve an approximation which means they produce biased samples from the target product (Vignac et al., 2022; Gruver et al., 2023; Nisonoff et al., 2024; Tang et al., 2025). Singhal et al. (2025) investigates the use of SMC to sample (in an asymptotically unbiased manner) from a reward-weighted distribution. Our work adapts such an unbiased SMC based strategy to a smoothly annealed form of the reward ($\beta_t r(x)$), and extends it to general products, and annealing. He et al. (2025) recently proposed another SMC-based technique for such problems, however, they do not evaluate the method on discrete diffusion tasks.

**Boltzmann distribution annealing.** Our approach for annealing is related to recent works which explore methods to train discrete neural samplers for combinatorial optimization and statistical physics. These include works such as scalable discrete diffusion samplers (Sanokowski et al., 2025), LEAPS (Holderrieth et al., 2025), and discrete neural flow samplers with locally equivariant transformers (Ou et al., 2025). These methods are typically trained to approximate Boltzmann distributions at a range of temperatures (or rely on access to the energy function during training), whereas our method assumes access to a trained model (perhaps with samples from a single temperature) and then uses it to generate samples at different, unseen temperatures through a modified inference algorithm.

**Theoretical guarantees.** Establishing the convergence bounds on the distribution of the produced samples is a direction of independent interest. In this work, we provide the proof of the Feynman-Kac formula (Theorem 2.1), which establishes the convergence of the time-discretization scheme. However, the community has developed a range of theoretical tools (Benton et al., 2023; Ren et al., 2024; Le-Tuyet-Nhi et al., 2025) potentially allowing for a more accurate convergence analysis.

## 6 CONCLUSION

In this paper, we propose DISCRETE FEYNMAN-KAC CORRECTORS— a framework that allows for re-purposing discrete diffusion models at inference time without retraining them. In particular, our theoretical findings demonstrate that sampling from the annealed, product or reward-weighted distributions can be efficiently done by combining the learned probability ratios and running SMC algorithms. Our empirical study supports our derivations and demonstrates that the proposed approach is more effective for tasks such as sampling from lower temperature Ising models, using language models to solve programming problems or inferring parameters, and controlling generated protein sequences. For future work we leave the extension to joint continuous and discrete models, as well as procedures to combine the method with reward fine-tuning.

## 7 REPRODUCIBILITY STATEMENT

To facilitate reproducibility of our empirical results and algorithm, we have made our code publicly available at this link: `https://anonymous.4open.science/r/discrete_fkc-40B8/README.md`. We describe all mathematical and algorithmic details necessary to reproduce our results throughout this paper (e.g. Alg. 1).

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

# A METHOD OVERVIEW

## A.1 CORRESPONDENCE TO CONTINUOUS FEYNMAN-KAC CORRECTORS

Table A1: Comparison of state updates for DISCRETE FEYNMAN-KAC CORRECTORS and continuous Feynman-Kac Correctors (Skreta et al., 2025), corresponding to line 4 in Alg. 1. For computing $r_{l,j}^{\text{diff}}$ in the case of sampling from the reward-tilted distribution, $x_t^{l,j}$ is $x_t$, except that position $l$ is replaced with token $j$, where $l$ is the position being de-masked and $j$ is a token from the vocabulary.

| | Target | DFKC | FKC |
|---|---|---|---|
| Base | $p_t(x)$ | $x_{t+\Delta t} \sim \texttt{softmax}(\text{NN}(x_t))$ | $x_{t+\Delta t} = x_t + (-f_t(x_t) + \sigma_t^2 \text{NN}(x_t))\Delta t + \sigma_t \Delta W_t$ |
| Annealing | $p_t^\beta(x)$ | $x_{t+\Delta t} \sim \texttt{softmax}(\beta \text{NN}(x_t))$ | $x_{t+\Delta t} = x_t + (-f_t(x_t) + \beta \sigma_t^2 \text{NN}(x_t))\Delta t + \sigma_t \Delta W_t$ |
| Product | $p_t^1(x)p_t^2(x)$ | $x_{t+\Delta t} \sim \texttt{softmax}(\text{NN}^1(x_t) + \text{NN}^2(x_t))$ | $x_{t+\Delta t} = x_t + (-f_t(x_t) + \sigma_t^2(\text{NN}^1(x_t) + \text{NN}^2(x_t)))\Delta t + \sigma_t \Delta W_t$ |
| Reward | $p_t(x)\exp(\beta_t r(x))$ | [1] $r_{l,j}^{\text{diff}} = \beta_t(r(x_t^{l,j}) - r(x_t))$ [2] $x_{t+\Delta t} \sim \texttt{softmax}(\text{NN}(x_t) + r^{\text{diff}})$ | $x_{t+\Delta t} = x_t + (-f_t(x_t) + \sigma_t^2 \text{NN}(x_t) + \beta_t \frac{\sigma_t^2}{2} \nabla r(x_t))\Delta t + \sigma_t \Delta W_t$ |

Table A2: Comparison of weight updates for DISCRETE FEYNMAN-KAC CORRECTORS and continuous Feynman-Kac Correctors (Skreta et al., 2025), corresponding to line 5 in Alg. 1.

| | Target | DFKC | FKC |
|---|---|---|---|
| Base | $p_t(x)$ | — | — |
| Annealing | $p_t^\beta(x)$ | $g_t(x_t) = \beta \frac{(1-t)^{\beta-1}}{t^\beta} \sum_j \texttt{softmax}(\beta \text{NN}(x_t)_j)$ | $g_t(x_t) = (\beta - 1)(\langle \nabla, f_t(x_t) \rangle + \frac{\sigma_t^2}{2}\beta\|\text{NN}(x_t)\|^2 \Delta t$ |
| Product | $p_t^1(x)p_t^2(x)$ | $g_t(x_t) = 2\frac{(1-t)}{t^2} \sum_j \texttt{softmax}(\text{NN}^1(x_t)_j + \text{NN}^2(x_t)_j)$ | $g_t(x_t) = (\langle \nabla, f_t(x_t) \rangle + \sigma_t^2 \langle \text{NN}^1(x_t), \text{NN}^2(x_t) \rangle)\Delta t$ |
| Reward | $p_t(x)\exp(\beta_t r(x))$ | $g_t(x_t) = \frac{1}{t} \sum_j \texttt{softmax}(\text{NN}(x_t)_j + r^{\text{diff}}) + \Delta \beta_t r(x_t)$ | $g_t(x_t) = (\langle \beta_t \nabla r(x_t), \frac{\sigma_t^2}{2} \text{NN}(x_t) - f_t(x_t) \rangle + \Delta \beta_t r(x_t))\Delta t$ |

# B  BACKGROUND PROOFS

## B.1  WEIGHTED FORWARD KOLMOGOROV EQUATION

Consider the forward Kolmogorov equation with the weighting term

$$\frac{\partial p_s(j)}{\partial s} = \sum_{k \neq j} A_s(k, j)p_s(k) - \sum_{k \neq j} A_s(j, k)p_s(j) + p_s(j)(g_s(j) - \sum_k p_s(k)g_s(k)). \tag{22}$$

We can re-write the last term as

$$p_s(j)(g_s(j) - \sum_k p_s(k)g_s(k)) = \sum_k p_s(k)p_s(j)(g_s(j) - g_s(k)) \tag{23}$$

$$= \sum_k p_s(k)p_s(j)\sigma_s(j, k)|g_s(j) - g_s(k)| \tag{24}$$

$$= \sum_k p_s(j)\mathbb{1}[\sigma_s(j, k) > 0]|g_s(j) - g_s(k)|p_s(k) - \tag{25}$$

$$- \sum_k p_s(k)\mathbb{1}[\sigma_s(j, k) < 0]|g_s(j) - g_s(k)|p_s(j), \tag{26}$$

where $\sigma_s(j, k)$ is the sign of $(g_s(j) - g_s(k))$. Let's define

$$B_s(k, j) := p_s(k)\mathbb{1}[\sigma_s(j, k) > 0]|g_s(j) - g_s(k)| \tag{27}$$

$$\implies B_s(j, k) := p_s(j)\mathbb{1}[\sigma_s(k, j) > 0]|g_s(k) - g_s(j)|. \tag{28}$$

Using the fact that $\sigma_s(k, j) = -\sigma_s(j, k)$, we have

$$p_s(j)(g_s(j) - \sum_k p_s(k)g_s(k)) = \sum_k B_s(k, j)p_s(k) - \sum_k B_s(j, k)p_s(j). \tag{29}$$

Finally, using the fact that $B_s(j, j) = 0$, we have

$$\frac{\partial p_s(j)}{\partial s} = \sum_{k \neq j} A_s(k, j)p_s(k) - \sum_{k \neq j} A_s(j, k)p_s(j) + p_s(j)(g_s(j) - \sum_k p_s(k)g_s(k)) \tag{30}$$

$$= \sum_{k \neq j}(A_s(k, j) + B_s(k, j))p_s(k) - \sum_{k \neq j}(A_s(j, k) + B_s(j, k))p_s(j), \tag{31}$$

$$B_s(k, j) := p_s(k)\mathbb{1}[\sigma_s(j, k) > 0]|g_s(j) - g_s(k)|. \tag{32}$$

## B.2  DISCRETE FEYNMAN-KAC FORMULA

**Theorem 2.1.** [Feynman-Kac Formula] *For the forward Kolmogorov equation from Equation* (4) *describing the time-evolution of the marginals $p_t(i)$ with the rate matrix $A_t(i, j)$ and weights $g_t(i)$, $\bar{g}_t(i) = g_t(i) - \sum_k p_t(k)g_t(k)$*

$$\mathbb{E}_{p_T(x)}\phi(x) = \lim_{dt \to 0} \sum_{x_T} \dots \sum_{x_0} \phi(x_T)p(x_T \mid x_{T-dt}) \dots p(x_{dt} \mid x_0) \exp\left(\sum_{t=0}^T dt\bar{g}_t(x_t)\right)p_0(x_0)$$

$$= \mathbb{E}_{X_{0:T}} \exp\left(\int_0^T dt\, \bar{g}_t(X_t)\right)\phi(X_T) \propto \mathbb{E}_{X_{0:T}} \exp\left(\int_0^T dt\, g_t(X_t)\right)\phi(X_T), \tag{5}$$

*where the expectation on the right hand side is taken w.r.t. trajectories $X_{0:T}$ defined as the limit of the transitions from Equation* (3).

*Proof.* We re-write the following FKE in the matrix notation, i.e.

$$\frac{\partial p_t(i)}{\partial t} = \sum_{j \neq i} A_t(j, i) p_t(j) - \sum_{j \neq i} A_t(i, j) p_t(i) + p_t(i) \Big( g_t(i) - \sum_k p_t(k) g_t(k) \Big) \tag{33}$$

$$= \sum_j A_t(j, i) p_t(j) + \sum_j \delta_{ij} \bar{g}_t(i) p_t(j) \tag{34}$$

$$= [\boldsymbol{A}_t \boldsymbol{p}_t]_i + [\boldsymbol{G}_t \boldsymbol{p}_t]_i, \tag{35}$$

where we define the matrices $\boldsymbol{A}_t$ and $\boldsymbol{G}_t$ as

$$[\boldsymbol{A}_t]_{ij} := A_t(j, i), \quad [\boldsymbol{G}_t]_{ij} := \delta_{ij} \bar{g}_t(i), \quad \bar{g}_t(i) = g_t(i) - \sum_k p_t(k) g_t(k). \tag{36}$$

Thus, in the matrix notation, we have the following Ordinary Differential Equation (ODE)

$$\frac{\partial \boldsymbol{p}_t}{\partial t} = (\boldsymbol{A}_t + \boldsymbol{G}_t) \boldsymbol{p}_t, \tag{37}$$

which solution is given by the time-ordered exponential, denoted as

$$\boldsymbol{p}_T = \mathcal{T} \exp\left( \int_0^T dt \, (\boldsymbol{A}_t + \boldsymbol{G}_t) \right) \boldsymbol{p}_0, \tag{38}$$

and defined as the following limit

$$\boldsymbol{p}_T = \lim_{n \to \infty} \prod_{k=0}^{n-1} \exp\left( \frac{1}{n} (\boldsymbol{A}_{(kT)/n} + \boldsymbol{G}_{(kT)/n}) \right) \boldsymbol{p}_0. \tag{39}$$

Using the time-dependent analog of the Lie-Trotter formula (see (Vuillermot, 2010) for the proof), we can re-write the matrix exponential of the sum as the product of matrix exponentials, which is not true in general because $\boldsymbol{A}_t$ and $\boldsymbol{G}_t$ do not commute i.e.

$$\lim_{n \to \infty} \prod_{k=0}^{n-1} \exp\left( \frac{1}{n} (\boldsymbol{A}_{(kT)/n} + \boldsymbol{G}_{(kT)/n}) \right) = \lim_{n \to \infty} \prod_{k=0}^{n-1} \exp\left( \frac{1}{n} \boldsymbol{A}_{(kT)/n} \right) \exp\left( \frac{1}{n} \boldsymbol{G}_{(kT)/n} \right).$$

Denoting $dt := 1/n$ and $t_k = (kT)/n$, we have

$$\boldsymbol{p}_T = \lim_{dt \to 0} \prod_{k=0}^{n-1} \exp(dt \boldsymbol{A}_{t_k}) \exp(dt \boldsymbol{G}_{t_k}) \boldsymbol{p}_0 \tag{40}$$

$$= \lim_{dt \to 0} \prod_{k=1}^{n-1} \exp(dt \boldsymbol{A}_{t_k}) \exp(dt \boldsymbol{G}_{t_k}) \sum_{j_0} \exp(dt \boldsymbol{A}_{t_0})_{i_1 j_0} \sum_{i_0} \exp(dt \boldsymbol{G}_{t_0})_{j_0 i_0} p_0(i_0). \tag{41}$$

Using the fact that $\boldsymbol{G}_t$ is diagonal, we have

$$\exp(dt \boldsymbol{G}_t)_{ij} = \delta_{ij} \exp(dt \bar{g}_t(i)), \tag{42}$$

and, correspondingly,

$$p_T(i) = \lim_{dt \to 0} \prod_{k=1}^{n-1} \exp(dt\boldsymbol{A}_{t_k}) \exp(dt\boldsymbol{G}_{t_k}) \sum_{j_0} \sum_{i_0} \exp(dt\boldsymbol{A}_{t_0})_{i_1 j_0} \delta_{j_0 i_0} \exp(dt\bar{g}_{t_0}(j_0)) p_0(i_0)$$

$$= \lim_{dt \to 0} \prod_{k=1}^{n-1} \exp(dt\boldsymbol{A}_{t_k}) \exp(dt\boldsymbol{G}_{t_k}) \sum_{i_0} \exp(dt\boldsymbol{A}_{t_0})_{i_1 i_0} \exp(dt\bar{g}_{t_0}(i_0)) p_0(i_0) \tag{43}$$

$$\cdots \tag{44}$$

$$= \lim_{dt \to 0} \sum_{i_{n-2}} \cdots \sum_{i_1} \sum_{i_0} \exp(dt\boldsymbol{A}_{t_{n-1}})_{i i_{n-2}} \exp(dt\bar{g}_{t_{n-2}}(i_{n-2})) \cdots \cdot \tag{45}$$

$$\cdot \exp(dt\boldsymbol{A}_{t_0})_{i_1 i_0} \exp(dt\bar{g}_{t_0}(i_0)) p_0(i_0) \tag{46}$$

$$= \lim_{dt \to 0} \sum_{i_{n-2}} \cdots \sum_{i_0} \exp(dt\boldsymbol{A}_{t_{n-1}})_{i i_{n-2}} \cdots \exp(dt\boldsymbol{A}_{t_0})_{i_1 i_0} \exp\left(\sum_{k=0}^{n-2} dt\bar{g}_{t_k}(i_k)\right) p_0(i_0).$$

Finally, we denote

$$p(x_{t+dt} = i \mid x_t = j) := \exp(dt\boldsymbol{A}_t)_{ij} = \delta_{ij} + A_t(j, i) dt + o(dt). \tag{47}$$

In this notation, the expected value of the statistics $\phi$ can be evaluated as

$$\sum_x \phi(x) p_T(x) = \lim_{dt \to 0} \sum_{x_T} \cdots \sum_{x_0} \phi(x_T) p(x_T \mid x_{T-dt}) \ldots p(x_{dt} \mid x_0) \exp\left(\sum_{t=0}^{T} dt\bar{g}_t(x_t)\right) p_0(x_0)$$

$$= \mathbb{E}_{X_{0:T}} \exp\left(\int_0^T dt\, \bar{g}_t(X_t)\right) \phi(X_T) \propto \mathbb{E}_{X_{0:T}} \exp\left(\int_0^T dt\, g_t(X_t)\right) \phi(X_T), \tag{48}$$

where in the last two formulas we take the expectation w.r.t. the process $X_{0:T}$ defined as the limit of the transition distributions $p(x_{t+dt} = i \mid x_t = j)$. $\qquad\square$

## B.3 DISCRETE MASKED DIFFUSION

First, we consider general case, where $m$ is the mask state and $\alpha_{s,t}$ is the noise schedule, i.e. the noising process is defined as

$$p(x_s = j \mid x_t = i) = (1 - \bar{\alpha}_{s,t})\delta_{mj} + \bar{\alpha}_{s,t}\delta_{ij}. \tag{49}$$

Note that not every $\bar{\alpha}_{s,t}$ satisfies the master equation and we have to ensure that the following equality holds.

$$p(x_s = j \mid x_t = i) = \sum_k p(x_s = j \mid x_r = k) p(x_r = k \mid x_t = i) \tag{50}$$

$$(1 - \bar{\alpha}_{s,t})\delta_{mj} + \bar{\alpha}_{s,t}\delta_{ij} = \sum_k ((1 - \bar{\alpha}_{s,r})\delta_{mj} + \bar{\alpha}_{s,r}\delta_{kj})((1 - \bar{\alpha}_{r,t})\delta_{mk} + \bar{\alpha}_{r,t}\delta_{ik}) \tag{51}$$

$$(1 - \bar{\alpha}_{s,t})\delta_{mj} + \bar{\alpha}_{s,t}\delta_{ij} = (1 - \bar{\alpha}_{s,r})\delta_{mj}(\bar{\alpha}_{r,t} + (1 - \bar{\alpha}_{r,t})) + \bar{\alpha}_{s,r}((1 - \bar{\alpha}_{r,t})\delta_{mj} + \bar{\alpha}_{r,t}\delta_{ij})$$

$$(1 - \bar{\alpha}_{s,t})\delta_{mj} + \bar{\alpha}_{s,t}\delta_{ij} = ((1 - \bar{\alpha}_{s,r}) + \bar{\alpha}_{s,r}(1 - \bar{\alpha}_{r,t}))\delta_{mj} + \bar{\alpha}_{s,r}\bar{\alpha}_{r,t}\delta_{ij}. \tag{52}$$

Thus, the following relations must hold

$$1 - \bar{\alpha}_{s,t} = (1 - \bar{\alpha}_{s,r}) + \bar{\alpha}_{s,r}(1 - \bar{\alpha}_{r,t}), \quad \bar{\alpha}_{s,t} = \bar{\alpha}_{s,r}\bar{\alpha}_{r,t} \tag{53}$$

$$-\bar{\alpha}_{s,t} = -\bar{\alpha}_{r,t}\bar{\alpha}_{s,r}, \quad \bar{\alpha}_{s,t} = \bar{\alpha}_{s,r}\bar{\alpha}_{r,t}, \tag{54}$$

$$\bar{\alpha}_{s,t} = \bar{\alpha}_{r,t}\bar{\alpha}_{s,r}. \tag{55}$$

Thus, any function that satisfy the following equation works

$$\forall\, t \le r \le s, \quad \bar{\alpha}_{s,t} = \bar{\alpha}_{s,r}\bar{\alpha}_{r,t}. \tag{56}$$

Denoting $\alpha_s = \bar{\alpha}_{s,0}$, we have

$$\bar{\alpha}_{s,t} = \frac{\alpha_s}{\alpha_t}, \text{ and } p(x_s = j \mid x_t = i) = \left(1 - \frac{\alpha_s}{\alpha_t}\right)\delta_{mj} + \frac{\alpha_s}{\alpha_t}\delta_{ij}. \tag{57}$$

From here, the rate matrix of the noising process is

$$A_t(i,j) = \left.\frac{\partial p(x_s = j \mid x_t = i)}{\partial s}\right|_{s=t} = \frac{1}{\alpha_t}\frac{\partial \alpha_t}{\partial t}(\delta_{ij} - \delta_{mj}). \tag{58}$$

### B.4 REVERSE-TIME MASKED DIFFUSION

For the inverse time $\tau = 1 - t$, we flip the marginals $q_\tau(i) := p_{1-\tau}(i)$ and take the derivative w.r.t. $\tau$

$$\frac{\partial q_\tau(i)}{\partial \tau} = \frac{\partial p_{1-\tau}(i)}{\partial \tau} = -\left.\frac{\partial p_t(i)}{\partial t}\right|_{t=1-\tau} \tag{59}$$

$$= -\sum_{j\neq i}(A_{1-\tau}(j,i)p_{1-\tau}(j) - A_{1-\tau}(i,j)p_{1-\tau}(i)) \tag{60}$$

$$= \sum_{j\neq i}\left(A_{1-\tau}(i,j)\frac{p_{1-\tau}(i)}{q_\tau(j)}q_\tau(j) - A_{1-\tau}(j,i)\frac{p_{1-\tau}(j)}{q_\tau(i)}q_\tau(i)\right) \tag{61}$$

$$= \sum_{j\neq i}(B_\tau(j,i)q_\tau(j) - B_\tau(i,j)q_\tau(i)), \quad B_\tau(i,j) := A_{1-\tau}(j,i)\frac{p_{1-\tau}(j)}{p_{1-\tau}(i)}. \tag{62}$$

Note that here we define only the off-diagonal elements and the diagonal elements are

$$B_\tau(i,i) = -\sum_{j\neq i}B_\tau(i,j) = -\sum_{j\neq i}A_{1-\tau}(j,i)\frac{p_{1-\tau}(j)}{p_{1-\tau}(i)}. \tag{63}$$

In particular, for the masked diffusion, we have

$$B_\tau(i,j) = \frac{1}{\alpha_t}\frac{\partial \alpha_t}{\partial t}(\delta_{ij} - \delta_{mi})\frac{p_t(j)}{p_t(i)}, \quad i \neq j \tag{64}$$

$$= -\frac{1}{\alpha_t}\frac{\partial \alpha_t}{\partial t}\frac{p_t(j)}{p_t(m)}\delta_{mi}, \tag{65}$$

$$B_\tau(i,i) = -\sum_{j\neq i}B_\tau(i,j) = \frac{1}{\alpha_t}\frac{\partial \alpha_t}{\partial t}\frac{1 - p_t(m)}{p_t(m)}\delta_{mi}. \tag{66}$$

### B.5 DE-MASKING PARAMETERIZATION

Furthermore, analogously to the derivation from (Shi et al., 2024) (Appendix H.3), we have

$$\frac{p_t(j)}{p_t(m)} = \sum_i \frac{p_0(i)}{p_t(m)}p(x_t = j \mid x_0 = i) \tag{67}$$

$$= \sum_i \frac{p_0(i)p(x_t = m \mid x_0 = i)}{p_t(m)p(x_0 = i \mid x_t = m)}\frac{p(x_0 = i \mid x_t = m)}{p(x_t = m \mid x_0 = i)}p(x_t = j \mid x_0 = i) \tag{68}$$

$$= \sum_i \frac{p(x_0 = i \mid x_t = m)}{p(x_t = m \mid x_0 = i)}p(x_t = j \mid x_0 = i) \tag{69}$$

$$= \sum_i \frac{p(x_0 = i \mid x_t = m)}{(1 - \alpha_t) + \alpha_t\delta_{im}}((1 - \alpha_t)\delta_{mj} + \alpha_t\delta_{ij}) \tag{70}$$

$$= \frac{1}{1 - \alpha_t}\sum_i((1 - \alpha_t)\delta_{mj} + \alpha_t\delta_{ij})p(x_0 = i \mid x_t = m) \tag{71}$$

$$= \delta_{mj} + \frac{\alpha_t}{1 - \alpha_t}p(x_0 = j \mid x_t = m). \tag{72}$$

where we used the fact that $p(x_0 = m) = 0$.

### B.6 MULTIDIMENSIONAL CASE

For the multi-dimensional case, we consider the masking process applied independently to each coordinate, i.e.

$$p(x_s = [j_1 \ldots j_d] \,|\, x_t = [i_1 \ldots i_d]) = \prod_{k=1}^{d} p(x_s[k] = j_k \,|\, x_t[k] = i_k) \tag{73}$$

$$= \prod_{k=1}^{d} \left( \left(1 - \frac{\alpha_s}{\alpha_t}\right) \delta_{m j_k} + \frac{\alpha_s}{\alpha_t} \delta_{i_k j_k} \right), \tag{74}$$

which defines the following rate matrix

$$A_t([i_1 \ldots i_d], [j_1 \ldots j_d]) = \left. \frac{\partial p(x_s = [j_1 \ldots j_d] \,|\, x_t = [i_1 \ldots i_d])}{\partial s} \right|_{s=t} \tag{75}$$

$$= \sum_{k=1}^{d} \prod_{l \neq k} p(x_t[l] = j_l \,|\, x_t[l] = i_l) \left. \frac{\partial p(x_s[k] = j_k \,|\, x_t[k] = i_k)}{\partial s} \right|_{s=t} \tag{76}$$

$$= \frac{1}{\alpha_t} \frac{\partial \alpha_t}{\partial t} \sum_{k=1}^{d} \prod_{l \neq k} \delta_{j_l i_l} (\delta_{i_k j_k} - \delta_{m j_k}). \tag{77}$$

For the off-diagonal elements of the reverse-time matrix, we have

$$B_t([i_1 \ldots i_d], [j_1 \ldots j_d]) = A_t([j_1 \ldots j_d], [i_1 \ldots i_d]) \frac{p_t([j_1 \ldots j_d])}{p_t([i_1 \ldots i_d])} \tag{78}$$

$$= \frac{1}{\alpha_t} \frac{\partial \alpha_t}{\partial t} \frac{p_t([j_1 \ldots j_d])}{p_t([i_1 \ldots i_d])} \sum_{k=1}^{d} \prod_{l \neq k} \delta_{j_l i_l} (\delta_{i_k j_k} - \delta_{m i_k}) \tag{79}$$

$$= -\frac{1}{\alpha_t} \frac{\partial \alpha_t}{\partial t} \frac{p_t([j_1 \ldots j_d])}{p_t([i_1 \ldots i_d])} \sum_{k=1}^{d} \prod_{l \neq k} \delta_{j_l i_l} \delta_{m i_k}. \tag{80}$$

## C  DISCRETE FEYNMAN-KAC CORRECTORS PROOFS

### C.1  ANNEALING OF FKE

**Theorem 3.1.** [Temperature Annealing] *Consider the forward Kolmogorov equation from Equation* (2) *describing the time-evolution of the marginals $p_t(i)$ with the rate matrix $A_t(i, j)$. For the temperature annealed marginals $q_t(i) \propto p_t(i)^{\beta}$, the following equation holds*

$$\frac{\partial q_t(i)}{\partial t} = \sum_{j \neq i} \left( A_t^{\text{anneal}}(j, i) q_t(j) - A_t^{\text{anneal}}(i, j) q_t(i) \right) + q_t(i) \left( g_t(i) - \mathbb{E}_{q_t(j)} g_t(j) \right), \quad (12)$$

*where* $A_t^{\text{anneal}}(i, j) := \beta A_t(i, j) \frac{p_t^{1-\beta}(i)}{p_t^{1-\beta}(j)}$, $g_t(i) := \sum_{j \neq i} \left( A_t^{\text{anneal}}(i, j) - \beta A_t(i, j) \right)$. (13)

*Proof.* Consider the forward Kolmogorov equation for the given rate matrix $A_t(i, j)$

$$\frac{\partial p_t(i)}{\partial t} = \sum_{j \neq i} A_t(j, i) p_t(j) - \sum_{j \neq i} A_t(i, j) p_t(i) \tag{81}$$

$$\frac{\partial}{\partial t} \log p_t(i) = \sum_{j \neq i} A_t(j, i) \frac{p_t(j)}{p_t(i)} - \sum_{j \neq i} A_t(i, j) = \sum_{j \neq i} \left( A_t(j, i) \frac{p_t(j)}{p_t(i)} - A_t(i, j) \right). \tag{82}$$

Then the annealed target $q_t(i) := p_t^{\beta}(i)/Z_t$ follows

$$\frac{\partial}{\partial t} \log q_t(i) = \beta \frac{\partial}{\partial t} \log p_t(i) - \frac{\partial}{\partial t} \log Z_t \tag{83}$$

$$= \sum_{j \neq i} \left( \beta A_t(j, i) \frac{p_t(j)}{p_t(i)} - \beta A_t(i, j) \right) - \frac{\partial}{\partial t} \log Z_t \tag{84}$$

$$= \sum_{j \neq i} \left( \underbrace{\beta A_t(j, i) \frac{p_t^{1-\beta}(j)}{p_t^{1-\beta}(i)}}_{:= A_t^{\text{anneal}}(j, i)} \frac{q_t(j)}{q_t(i)} - A_t^{\text{anneal}}(i, j) \right) + \tag{85}$$

$$+ \sum_{j \neq i} \left( A_t^{\text{anneal}}(i, j) - \beta A_t(i, j) \right) - \frac{\partial}{\partial t} \log Z_t. \tag{86}$$

Denoting the second term as $g_t(j)$, we have

$$\frac{\partial q_t(i)}{\partial t} = \sum_{j \neq i} \left( A_t^{\text{anneal}}(j, i) q_t(j) - A_t^{\text{anneal}}(i, j) q_t(i) \right) + q_t(i) \left( g_t(i) - \frac{\partial}{\partial t} \log Z_t \right), \tag{87}$$

$$A_t^{\text{anneal}}(j, i) := \beta A_t(j, i) \frac{p_t^{1-\beta}(j)}{p_t^{1-\beta}(i)}, \quad g_t(i) := \sum_{j \neq i} \left( A_t^{\text{anneal}}(i, j) - \beta A_t(i, j) \right). \tag{88}$$

From the definition of $q_t(i)$ we have

$$\sum_i q_t(i) = 1, \quad \forall t, \tag{89}$$

hence,

$$\sum_i \frac{\partial q_t(i)}{\partial t} = 0 \implies \sum_i q_t(i) \left( g_t(i) - \frac{\partial}{\partial t} \log Z_t \right) = 0, \tag{90}$$

which immediately yields

$$g_t(i) - \frac{\partial}{\partial t} \log Z_t = g_t(i) - \mathbb{E}_{i \sim q_t(i)} g_t(i). \tag{91}$$

However, one can also verify this through the definition of the normalization constant

$$\frac{\partial}{\partial t} \log Z_t = \frac{1}{Z_t} \sum_i \frac{\partial p_t^\beta(i)}{\partial t} = \sum_i \frac{p_t^\beta(i)}{Z_t} \beta \frac{\partial}{\partial t} \log p_t(i) \tag{92}$$

$$= \sum_i q_t(i) \sum_{j \neq i} \left( \beta A_t(j,i) \frac{p_t(j)}{p_t(i)} - \beta A_t(i,j) \right), \tag{93}$$

and, correspondingly

$$\sum_i q_t(i) g_t(i) - \frac{\partial}{\partial t} \log Z_t = \sum_i q_t(i) \sum_{j \neq i} \left( \beta A_t(i,j) \frac{p_t^{1-\beta}(i)}{p_t^{1-\beta}(j)} - \beta A_t(j,i) \frac{p_t(j)}{p_t(i)} \right) \tag{94}$$

$$= \frac{\beta}{Z_t} \sum_i \sum_{j \neq i} \left( A_t(i,j) \frac{p_t(i)}{p_t^{1-\beta}(j)} - A_t(j,i) \frac{p_t(j)}{p_t^{1-\beta}(i)} \right) \tag{95}$$

$$= \frac{\beta}{Z_t} \left( \sum_i \sum_{j \neq i} \hat{A}_t(i,j) - \sum_i \sum_{j \neq i} \hat{A}_t(j,i) \right) \tag{96}$$

$$= \frac{\beta}{Z_t} \left( \sum_{i,j} \hat{A}_t(i,j) - \sum_{i,j} \hat{A}_t(j,i) \right) = 0, \tag{97}$$

where we denote $\hat{A}_t(i,j) := A_t(i,j) \frac{p_t(i)}{p_t^{1-\beta}(j)}$.

Thus, we have

$$\frac{\partial q_t(i)}{\partial t} = \sum_{j \neq i} \left( A_t^{\text{anneal}}(j,i) q_t(j) - A_t^{\text{anneal}}(i,j) q_t(i) \right) + q_t(i) \left( g_t(i) - \mathbb{E}_{q_t(j)} g_t(j) \right), \tag{98}$$

$$A_t^{\text{anneal}}(j,i) := \beta A_t(j,i) \frac{p_t^{1-\beta}(j)}{p_t^{1-\beta}(i)}, \quad g_t(i) := \sum_{j \neq i} \left( A_t^{\text{anneal}}(i,j) - \beta A_t(i,j) \right). \tag{99}$$

$\square$

**Corollary C.1.** [Annealed Masked Diffusion] *For the rate matrix of the reverse-time masked diffusion from Equation (10), Theorem 3.1 yields the following off-diagonal elements of the rate matrix and the corresponding weight function*

$$B_\tau^{\text{anneal}}(i,j) = -\delta_{mi} \frac{\beta}{\alpha_t} \frac{\partial \alpha_t}{\partial t} \frac{p_t^\beta(j)}{p_t^\beta(m)}, \quad g_\tau(i) = \delta_{mi} \frac{\beta}{\alpha_t} \frac{\partial \alpha_t}{\partial t} \sum_j \left( \frac{p_t(j)}{p_t(m)} - \frac{p_t^\beta(j)}{p_t^\beta(m)} \right). \tag{14}$$

*Proof.* The reverse-time rate matrix is

$$B_t(i,j) = -\delta_{mi} \frac{1}{\alpha_t} \frac{\partial \alpha_t}{\partial t} \frac{p_t(j)}{p_t(m)}, \quad i \neq j. \tag{100}$$

Then, according to Theorem 3.1, the rate matrix of the annealed process is

$$B_t^{\text{anneal}}(i,j) = \beta B_t(i,j) \frac{p_t^{1-\beta}(i)}{p_t^{1-\beta}(j)} = -\delta_{mi} \frac{\beta}{\alpha_t} \frac{\partial \alpha_t}{\partial t} \frac{p_t(j)}{p_t(m)} \frac{p_t^{1-\beta}(i)}{p_t^{1-\beta}(j)} = -\delta_{mi} \frac{\beta}{\alpha_t} \frac{\partial \alpha_t}{\partial t} \frac{p_t^\beta(j)}{p_t^\beta(m)} \tag{101}$$

And the weighting term is

$$g_t(i) = \sum_{j \neq i} \left( B_t^{\text{anneal}}(i,j) - \beta B_t(i,j) \right) = \delta_{mi} \frac{\beta}{\alpha_t} \frac{\partial \alpha_t}{\partial t} \sum_{j \neq i} \left( \frac{p_t(j)}{p_t(m)} - \frac{p_t^\beta(j)}{p_t^\beta(m)} \right) \tag{102}$$

$$= \delta_{mi} \frac{\beta}{\alpha_t} \frac{\partial \alpha_t}{\partial t} \sum_{j \neq m} \left( \frac{p_t(j)}{p_t(m)} - \frac{p_t^\beta(j)}{p_t^\beta(m)} \right) = \delta_{mi} \frac{\beta}{\alpha_t} \frac{\partial \alpha_t}{\partial t} \sum_{j} \left( \frac{p_t(j)}{p_t(m)} - \frac{p_t^\beta(j)}{p_t^\beta(m)} \right) \tag{103}$$

$$\square$$

## C.2  Product of FKEs

**Theorem 3.3.** [Product of FKEs] *Consider two forward Kolmogorov equations (from Equation (2)) with different rate matrices $A_t^1(i,j)$ and $A_t^2(i,j)$ describing the evolution of marginals $p_t^1(i)$ and $p_t^2(i)$. For the product of marginals $q_t(i) \propto p_t^1(i)p_t^2(i)$, the following equation holds*

$$\frac{\partial q_t(i)}{\partial t} = \sum_{j \neq i} \left( A_t^{\text{prod}}(j,i)q_t(j) - A_t^{\text{prod}}(i,j)q_t(i) \right) + q_t(i) \left( g_t(i) - \mathbb{E}_{j \sim q_t(j)} g_t(j) \right), \tag{16}$$

$$A_t^{\text{prod}}(i,j) := A_t^1(i,j)\frac{p_t^2(j)}{p_t^2(i)} + A_t^2(i,j)\frac{p_t^1(j)}{p_t^1(i)}, \; g_t(i) := \sum_{j \neq i} \left( A_t^{\text{prod}}(i,j) - A_t^1(i,j) - A_t^2(i,j) \right).$$

*Proof.* Consider two forward Kolmogorov equations with different rate matrices $A_t^1(i,j)$ and $A_t^2(i,j)$. For both we have the equations of the form

$$\frac{\partial p_t^{1,2}(i)}{\partial t} = \sum_{j \neq i} A_t^{1,2}(j,i)p_t^{1,2}(j) - \sum_{j \neq i} A_t^{1,2}(i,j)p_t^{1,2}(i) \tag{104}$$

$$\frac{\partial}{\partial t} \log p_t^{1,2}(i) = \sum_{j \neq i} A_t^{1,2}(j,i)\frac{p_t^{1,2}(j)}{p_t^{1,2}(i)} - \sum_{j \neq i} A_t^{1,2}(i,j) \tag{105}$$

$$= \sum_{j \neq i} \left( A_t^{1,2}(j,i)\frac{p_t^{1,2}(j)}{p_t^{1,2}(i)} - A_t^{1,2}(i,j) \right). \tag{106}$$

Correspondingly, for the density $q_t(i) := p_t^1(i)p_t^2(i)/Z_t$, we have

$$\frac{\partial}{\partial t} \log q_t(i) = \frac{\partial}{\partial t} \log p_t^1(i) + \frac{\partial}{\partial t} \log p_t^2(i) - \frac{\partial}{\partial t} \log Z_t \tag{107}$$

$$= \sum_{j \neq i} \left( A_t^1(j,i)\frac{p_t^1(j)}{p_t^1(i)} - A_t^1(i,j) + A_t^2(j,i)\frac{p_t^2(j)}{p_t^2(i)} - A_t^2(i,j) \right) - \frac{\partial}{\partial t} \log Z_t \tag{108}$$

$$= \sum_{j \neq i} \left( A_t^1(j,i)\frac{p_t^2(i)}{p_t^2(j)}\frac{q_t(j)}{q_t(i)} + A_t^2(j,i)\frac{p_t^1(i)}{p_t^1(j)}\frac{q_t(j)}{q_t(i)} - A_t^1(i,j) - A_t^2(i,j) \right) - \frac{\partial}{\partial t} \log Z_t \tag{109}$$

$$= \sum_{j \neq i} \left( \underbrace{\left[ A_t^1(j,i)\frac{p_t^2(i)}{p_t^2(j)} + A_t^2(j,i)\frac{p_t^1(i)}{p_t^1(j)} \right]}_{:=A_t^{\text{prod}}(j,i)} \frac{q_t(j)}{q_t(i)} - A_t^1(i,j) - A_t^2(i,j) \right) - \frac{\partial}{\partial t} \log Z_t \tag{110}$$

$$= \sum_{j \neq i} \left( A_t^{\text{prod}}(j,i)\frac{q_t(j)}{q_t(i)} - A_t^{\text{prod}}(i,j) \right) + \tag{111}$$

$$+ \underbrace{\sum_{j \neq i} \left( A_t^{\text{prod}}(i,j) - A_t^1(i,j) - A_t^2(i,j) \right)}_{:=g_t(i)} - \frac{\partial}{\partial t} \log Z_t. \tag{112}$$

Finally, we have to show that the weights are self-normalized, i.e.

$$g_t(i) - \frac{\partial}{\partial t} \log Z_t = g_t(i) - \mathbb{E}_{i \sim q_t(j)} g_t(j). \tag{113}$$

Expanding the derivative of the normalization constant, we have

$$\frac{\partial}{\partial t} \log Z_t = \frac{1}{Z_t} \sum_i \left( p_t^1(i) \frac{\partial p_t^2(i)}{\partial t} + p_t^2(i) \frac{\partial p_t^1(i)}{\partial t} \right) = \sum_i q_t(i) \left( \frac{\partial}{\partial t} \log p_t^2(i) + \frac{\partial}{\partial t} \log p_t^1(i) \right)$$

$$= \sum_i q_t(i) \sum_{j \neq i} \left( A_t^1(j,i) \frac{p_t^1(j)}{p_t^1(i)} - A_t^1(i,j) + A_t^2(j,i) \frac{p_t^2(j)}{p_t^2(i)} - A_t^2(i,j) \right). \tag{114}$$

Thus, we have

$$\sum_i q_t(i) g_t(i) - \frac{\partial}{\partial t} \log Z_t = \sum_i q_t(i) \sum_{j \neq i} \left( A_t^{\text{prod}}(i,j) - A_t^1(j,i) \frac{p_t^1(j)}{p_t^1(i)} - A_t^2(j,i) \frac{p_t^2(j)}{p_t^2(i)} \right)$$

$$= \sum_i q_t(i) \sum_{j \neq i} \left( A_t^1(i,j) \frac{p_t^2(j)}{p_t^2(i)} + A_t^2(i,j) \frac{p_t^1(j)}{p_t^1(i)} - A_t^1(j,i) \frac{p_t^1(j)}{p_t^1(i)} - A_t^2(j,i) \frac{p_t^2(j)}{p_t^2(i)} \right) \tag{115}$$

$$= \frac{1}{Z_t} \sum_i \sum_{j \neq i} \left( A_t^1(i,j) p_t^1(i) p_t^2(j) + A_t^2(i,j) p_t^1(j) p_t^2(i) - \right. \tag{116}$$

$$\left. - A_t^1(j,i) p_t^1(j) p_t^2(i) - A_t^2(j,i) p_t^1(i) p_t^2(j) \right). \tag{117}$$

Denoting

$$\hat{A}_t(i,j) := A_t^1(i,j) p_t^1(i) p_t^2(j) + A_t^2(i,j) p_t^1(j) p_t^2(i), \tag{118}$$

we can show

$$\sum_i q_t(i) g_t(i) - \frac{\partial}{\partial t} \log Z_t = \frac{1}{Z_t} \sum_i \sum_{j \neq i} \left( \hat{A}_t(i,j) - \hat{A}_t(j,i) \right) \tag{119}$$

$$= \frac{1}{Z_t} \sum_{i,j} \left( \hat{A}_t(i,j) - \hat{A}_t(j,i) \right) = 0. \tag{120}$$

Thus, we have the result of the theorem, i.e.

$$\frac{\partial q_t(i)}{\partial t} = \sum_{j \neq i} \left( A_t^{\text{prod}}(j,i) \, q_t(j) - A_t^{\text{prod}}(i,j) q_t(i) \right) + q_t(i) \left( g_t(i) - \mathbb{E}_{j \sim q_t(j)} g_t(j) \right), \tag{121}$$

$$\text{where } A_t^{\text{prod}}(i,j) := A_t^1(i,j) \frac{p_t^2(j)}{p_t^2(i)} + A_t^2(i,j) \frac{p_t^1(j)}{p_t^1(i)}, \tag{122}$$

$$g_t(i) := \sum_{j \neq i} \left( A_t^{\text{prod}}(i,j) - A_t^1(i,j) - A_t^2(i,j) \right). \tag{123}$$

$\square$

**Corollary C.2.** [Product of Masked Diffusions] *For the rate matrix of the reverse-time masked diffusion from Equation* (10), *Theorem 3.3 yields*

$$B_\tau^{\text{prod}}(i,j) = -2\delta_{mi} \frac{1}{\alpha_t} \frac{\partial \alpha_t}{\partial t} \frac{p_t^1(j)}{p_t^1(m)} \frac{p_t^2(j)}{p_t^2(m)}, g_\tau(i) = \frac{\delta_{mi}}{\alpha_t} \frac{\partial \alpha_t}{\partial t} \sum_j \frac{p_t^1(j)}{p_t^1(m)} + \frac{p_t^2(j)}{p_t^2(m)} - 2\frac{p_t^1(j)}{p_t^1(m)} \frac{p_t^2(j)}{p_t^2(m)}$$

*Proof.* The reverse-time rate matrices are

$$B_t^1(i,j) = -\delta_{mi} \frac{1}{\alpha_t} \frac{\partial \alpha_t}{\partial t} \frac{p_t^1(j)}{p_t^1(m)}, \quad B_t^2(i,j) = -\delta_{mi} \frac{1}{\alpha_t} \frac{\partial \alpha_t}{\partial t} \frac{p_t^2(j)}{p_t^2(m)}. \tag{124}$$

Then, according to Theorem 3.3, the rate matrix for the product is

$$B_t^{\text{prod}}(i,j) = B_t^1(i,j)\frac{p_t^2(j)}{p_t^2(i)} + B_t^2(i,j)\frac{p_t^1(j)}{p_t^1(i)} \tag{125}$$

$$= -\delta_{mi}\frac{1}{\alpha_t}\frac{\partial\alpha_t}{\partial t}\frac{p_t^1(j)}{p_t^1(m)}\frac{p_t^2(j)}{p_t^2(i)} - \delta_{mi}\frac{1}{\alpha_t}\frac{\partial\alpha_t}{\partial t}\frac{p_t^2(j)}{p_t^2(m)}\frac{p_t^1(j)}{p_t^1(i)} \tag{126}$$

$$= -\delta_{mi}\frac{2}{\alpha_t}\frac{\partial\alpha_t}{\partial t}\frac{p_t^1(j)}{p_t^1(m)}\frac{p_t^2(j)}{p_t^2(m)} \,. \tag{127}$$

And the weighting term is

$$g_t(i) = \sum_{j\neq i}\left(B_t^{\text{prod}}(i,j) - B_t^1(i,j) - B_t^2(i,j)\right) \tag{128}$$

$$= \delta_{mi}\frac{1}{\alpha_t}\frac{\partial\alpha_t}{\partial t}\sum_j\left(\frac{p_t^1(j)}{p_t^1(m)} + \frac{p_t^2(j)}{p_t^2(m)} - 2\frac{p_t^1(j)}{p_t^1(m)}\frac{p_t^2(j)}{p_t^2(m)}\right). \tag{129}$$

$$\square$$

### C.3 GEOMETRIC AVERAGE OF FKES

**Theorem C.3.** [Geometric Average of FKEs] *Consider $N$ forward Kolmogorov equations with marginals $p_t^n(i)$ and corresponding rate matrices $A_t^n(i,j)$. For the geometric average of marginals $q_t(i) \propto \prod_{n=1}^N p_t^n(i)^{\beta_n}$, with $\sum_{i=1}^N \beta_n = 1$, the following equation holds*

$$\frac{\partial q_t(i)}{\partial t} = \sum_{j\neq i}\left(A_t^{\text{geom}}(j,i)\,q_t(j) - A_t^{\text{geom}}(i,j)q_t(i)\right) + q_t(i)\big(g_t(i) - \mathbb{E}_{q_t(j)}g_t(j)\big), \tag{130}$$

$$\text{where } A_t^{\text{geo}}(i,j) := \prod_{n=1}^N\left(\frac{p_t^n(j)}{p_t^n(i)}\right)^{\beta_n}\sum_{n=1}^N\beta_n A_t^n(j,i)\frac{p_t^n(j)}{p_t^n(i)}, \tag{131}$$

$$g_t(i) := \sum_{j\neq i}\left(A_t^{\text{geo}}(i,j) - \sum_{n=1}^N\beta_n A_t^n(i,j)\right). \tag{132}$$

*Proof.* We define the target marginals as

$$q_t(i) := \frac{1}{Z_t}\prod_{n=1}^N p_t^n(i)^{\beta_n}\,, \quad Z_t = \sum_i\prod_{n=1}^N p_t^n(i)^{\beta_n}\,. \tag{133}$$

Hence, the time derivative of the marginals is

$$\frac{\partial}{\partial t}\log q_t(i) = \sum_{n=1}^N\beta_n\frac{\partial}{\partial t}\log p_t^n(i) - \frac{\partial}{\partial t}\log Z_t \tag{134}$$

$$= \sum_{j\neq i}\sum_{n=1}^N\beta_n\left(A_t^n(j,i)\frac{p_t^n(j)}{p_t^n(i)} - A_t^n(i,j)\right) - \frac{\partial}{\partial t}\log Z_t \tag{135}$$

$$= \sum_{j\neq i}\left(\underbrace{\sum_{n=1}^N\beta_n A_t^n(j,i)\frac{p_t^n(j)}{p_t^n(i)}\frac{q_t(i)}{q_t(j)}}_{:=A_t^{\text{geom}}(j,i)}\frac{q_t(j)}{q_t(i)} - A_t^{\text{geom}}(i,j)\right) + \tag{136}$$

$$+ \sum_{j\neq i}\left(A_t^{\text{geom}}(i,j) - \sum_{n=1}^N\beta_n A_t^n(i,j)\right) - \frac{\partial}{\partial t}\log Z_t\,. \tag{137}$$

Denoting

$$A_t^{\text{geom}}(i,j) := \prod_{n=1}^{N} \left( \frac{p_t^n(j)}{p_t^n(i)} \right)^{\beta_n} \sum_{n=1}^{N} \beta_n A_t^n(i,j) \frac{p_t^n(i)}{p_t^n(j)} \, , \text{ and} \tag{138}$$

$$g_t(i) := \sum_{j \neq i} \left( A_t^{\text{geom}}(i,j) - \sum_{n=1}^{N} \beta_n A_t^n(i,j) \right), \tag{139}$$

we can describe the evolution of the marginals $q_t(i)$ as

$$\frac{\partial q_t(i)}{\partial t} = \sum_{j \neq i} (A_t^{\text{geom}}(j,i) q_t(j) - A_t^{\text{geom}}(i,j) q_t(i)) + q_t(i) \big( g_t(i) - \mathbb{E}_{j \sim q_t(j)} g_t(j) \big). \tag{140}$$

$\square$

**Corollary C.4.** [Geometric Average of Masked Diffusions] *For the rate matrix of the reverse-time masked diffusion from Equation* (10)*, Theorem C.3 yields*

$$B_t^{\text{geom}}(i,j) = -\delta_{mi} \frac{1}{\alpha_t} \frac{\partial \alpha_t}{\partial t} \prod_{n=1}^{N} \left( \frac{p_t^n(j)}{p_t^n(m)} \right)^{\beta_n} \, , \, i \neq j \tag{141}$$

$$g_t(i) = \delta_{mi} \frac{1}{\alpha_t} \frac{\partial \alpha_t}{\partial t} \sum_{j \neq i} \left( \sum_{n=1}^{N} \beta_n \frac{p_t^n(j)}{p_t^n(m)} - \prod_{n=1}^{N} \left( \frac{p_t^n(j)}{p_t^n(m)} \right)^{\beta_n} \right). \tag{142}$$

*Proof.* For the reverse-time masked diffusion, we have

$$B_t^n(i,j) = -\delta_{mi} \frac{1}{\alpha_t} \frac{\partial \alpha_t}{\partial t} \frac{p_t^n(j)}{p_t^n(m)} \, , \, i \neq j \, , \, n = 1, \ldots, N \, . \tag{143}$$

Using the result of Theorem C.3, we have

$$B_t^{\text{geom}}(i,j) = -\delta_{mi} \frac{1}{\alpha_t} \frac{\partial \alpha_t}{\partial t} \prod_{n=1}^{N} \left( \frac{p_t^n(j)}{p_t^n(i)} \right)^{\beta_n} \sum_{n=1}^{N} \beta_n B_t^n(i,j) \frac{p_t^n(i)}{p_t^n(j)} \tag{144}$$

$$= -\delta_{mi} \frac{1}{\alpha_t} \frac{\partial \alpha_t}{\partial t} \prod_{n=1}^{N} \left( \frac{p_t^n(j)}{p_t^n(i)} \right)^{\beta_n} \sum_{n=1}^{N} \beta_n \frac{p_t^n(j)}{p_t^n(m)} \frac{p_t^n(i)}{p_t^n(j)} \tag{145}$$

$$= -\delta_{mi} \frac{1}{\alpha_t} \frac{\partial \alpha_t}{\partial t} \prod_{n=1}^{N} \left( \frac{p_t^n(j)}{p_t^n(m)} \right)^{\beta_n} \, , \tag{146}$$

where in the last transition we have used the fact that the expression is zero unless $i = m$ and $\sum_{n=1}^{N} \beta_n = 1$. Correspondingly, the weights are

$$g_t(i) = \sum_{j \neq i} \left( B_t^{\text{geom}}(i,j) - \sum_{n=1}^{N} \beta_n B_t^n(i,j) \right) \tag{147}$$

$$= \delta_{mi} \frac{1}{\alpha_t} \frac{\partial \alpha_t}{\partial t} \sum_{j \neq i} \left( \sum_{n=1}^{N} \beta_n \frac{p_t^n(j)}{p_t^n(m)} - \prod_{n=1}^{N} \left( \frac{p_t^n(j)}{p_t^n(m)} \right)^{\beta_n} \right). \tag{148}$$

$\square$

## C.4 REWARD-TILTED FKE

**Theorem 3.5.** [Reward-tilted FKE] *Consider the forward Kolmogorov equation from Equation (2) describing the time evolution of the marginals $p_t(i)$ with the rate matrix $A_t(i,j)$. For the reward-tilted marginals $q_t(i) \propto p_t(i) \exp(\beta_t r(i))$, the following equation holds*

$$\frac{\partial q_t(i)}{\partial t} = \sum_{j \neq i} \left( A_t^{\mathrm{reward}}(j,i) q_t(j) - A_t^{\mathrm{reward}}(i,j) q_t(i) \right) + q_t(i) \big( g_t(i) - \mathbb{E}_{q_t(j)} g_t(j) \big) , \quad (17)$$

$$A_t^{\mathrm{reward}}(i,j) := A_t(i,j) \frac{\exp(\beta_t r(j))}{\exp(\beta_t r(i))} , \quad g_t(i) := \sum_{j \neq i} \left( A_t^{\mathrm{reward}}(i,j) - A_t(i,j) \right) + \frac{\partial \beta_t}{\partial t} r(i) . \quad (18)$$

*Proof.* We define

$$q_t(i) := \frac{1}{Z_t} p_t(i) \exp(\beta_t r(i)) , \quad Z_t = \sum_i p_t(i) \exp(\beta_t r(i)) \quad (149)$$

The derivative of the log-probability is

$$\frac{\partial}{\partial t} \log q_t(i) = \sum_{j \neq i} \left( A_t(j,i) \frac{p_t(j)}{p_t(i)} - A_t(i,j) \right) + \frac{\partial \beta_t}{\partial t} r(i) - \frac{\partial}{\partial t} \log Z_t \quad (150)$$

$$= \sum_{j \neq i} \left( \underbrace{A_t(j,i) \frac{\exp(\beta_t r(i))}{\exp(\beta_t r(j))}}_{:=A_t^{\mathrm{reward}}(j,i)} \frac{q_t(j)}{q_t(i)} - A_t(i,j) \right) + \frac{\partial \beta_t}{\partial t} r(i) - \frac{\partial}{\partial t} \log Z_t \quad (151)$$

$$= \sum_{j \neq i} \left( A_t^{\mathrm{reward}}(j,i) \frac{q_t(j)}{q_t(i)} - A_t^{\mathrm{reward}}(i,j) \right) + \quad (152)$$

$$+ \underbrace{\sum_{j \neq i} (A_t^{\mathrm{reward}}(i,j) - A_t(i,j)) + \frac{\partial \beta_t}{\partial t} r(i)}_{:=g_t(i)} - \frac{\partial}{\partial t} \log Z_t \quad (153)$$

To show the following equality

$$g_t(i) - \frac{\partial}{\partial t} \log Z_t = g_t(i) - \mathbb{E}_{i \sim q_t(j)} g_t(j) , \quad (154)$$

one can either use the definition of $q_t(i)$ and its normalization, or explicitly calculate the derivative of the normalizing constant, i.e.

$$\frac{\partial}{\partial t} \log Z_t = \frac{1}{Z_t} \sum_i \frac{\partial}{\partial t} \Big( p_t(i) \exp(\beta_t r(i)) \Big) \quad (155)$$

$$= \sum_i q_t(i) \Big( \frac{\partial}{\partial t} \log p_t(i) + \frac{\partial \beta_t}{\partial t} r(i) \Big) \quad (156)$$

$$= \sum_i q_t(i) \Big( \sum_{j \neq i} \Big( A_t(j,i) \frac{p_t(j)}{p_t(i)} - A_t(i,j) \Big) + \frac{\partial \beta_t}{\partial t} r(i) \Big) \quad (157)$$

Thus, we have

$$\sum_i q_t(i)\, g_t(i) - \frac{\partial}{\partial t}\log Z_t = \sum_i q_t(i)\Big(\Big(\sum_{j\neq i} A_t^{\text{reward}}(i,j) - A_t(i,j)\Big) + \frac{\partial\beta_t}{\partial t}r(i) \tag{158}$$

$$- \big(\sum_{j\neq i} A_t(j,i)\frac{p_t(j)}{p_t(i)} - A_t(i,j)\big) - \frac{\partial\beta_t}{\partial t}r(i)\Big) \tag{159}$$

$$= \sum_i q_t(i)\sum_{j\neq i}\Big(A_t^{\text{reward}}(i,j) - A_t(j,i)\frac{p_t(j)}{p_t(i)}\Big) \tag{160}$$

$$= \sum_i q_t(i)\sum_{j\neq i}\Big(A_t(i,j)\frac{\exp(\beta_t r(j))}{\exp(\beta_t r(i))} - A_t(j,i)\frac{p_t(j)}{p_t(i)}\Big) \tag{161}$$

$$= \frac{1}{Z_t}\sum_i\sum_{j\neq i}\Big(A_t(i,j)\exp(\beta_t r(j))p_t(i) - A_t(j,i)\exp(\beta_t r(i))p_t(j)\Big) \tag{162}$$

$$= \frac{1}{Z_t}\sum_i\sum_{j\neq i}\Big(\hat{A}_t(i,j) - \hat{A}_t(j,i)\Big) = \frac{1}{Z_t}\sum_{i,j}\Big(\hat{A}_t(i,j) - \hat{A}_t(j,i)\Big) = 0\,, \tag{163}$$

where we denote

$$\hat{A}_t(i,j) := A_t(i,j)\exp(\beta_t r(j))p_t(i)\,. \tag{164}$$

Finally, we have

$$\frac{\partial q_t(i)}{\partial t} = \sum_{j\neq i}\big(A_t^{\text{reward}}(j,i)q_t(j) - A_t^{\text{reward}}(i,j)q_t(i)\big) + q_t(i)\big(g_t(i) - \mathbb{E}_{j\sim q_t(j)}g_t(j)\big)\,, \tag{165}$$

$$A_t^{\text{reward}}(i,j) := A_t(i,j)\frac{\exp(\beta_t r(j))}{\exp(\beta_t r(i))}\,, \quad g_t(i) := \sum_{j\neq i}\Big(A_t^{\text{reward}}(i,j) - A_t(i,j)\Big) + \frac{\partial\beta_t}{\partial t}r(i)\,.$$

$$\square$$

**Corollary C.5.** [Reward-tilted Masked Diffusion] *For the rate matrix of the reverse-time masked diffusion from Equation* (10)*, Theorem 3.5 yields*

$$B_\tau^{\text{reward}}(i,j) = -\delta_{mi}\frac{1}{\alpha_t}\frac{\partial\alpha_t}{\partial t}\frac{p_t(j)}{p_t(m)}\frac{\exp(\beta_t r(j))}{\exp(\beta_t r(m))}\,, \tag{19}$$

$$g_\tau(i) = \frac{1}{\alpha_t}\frac{\partial\alpha_t}{\partial t}\delta_{mi}\sum_j\Big(\frac{p_t(j)}{p_t(m)} - \frac{p_t(j)}{p_t(m)}\frac{\exp(\beta_t r(j))}{\exp(\beta_t r(m))}\Big) + \frac{\partial\beta_t}{\partial t}r(i)\,. \tag{20}$$

*Proof.* The reverse-time rate matrix is

$$B_t(i,j) = -\delta_{mi}\frac{1}{\alpha_t}\frac{\partial\alpha_t}{\partial t}\frac{p_t(j)}{p_t(m)}\,. \tag{166}$$

Then the reward-weighted matrix is

$$B_t^{\text{reward}}(i,j) = B_t(i,j)\frac{\exp(\beta_t r(j))}{\exp(\beta_t r(i))} = -\delta_{mi}\frac{1}{\alpha_t}\frac{\partial\alpha_t}{\partial t}\frac{p_t(j)}{p_t(m)}\frac{\exp(\beta_t r(j))}{\exp(\beta_t r(m))}\,, \tag{167}$$

and the weighting term is

$$g_t(i) = \sum_{j\neq i}\big(B_t^{\text{reward}}(i,j) - B_t(i,j)\big) + \frac{\partial\beta_t}{\partial t}r(i) \tag{168}$$

$$= \delta_{mi}\frac{1}{\alpha_t}\frac{\partial\alpha_t}{\partial t}\sum_j\Big(\frac{p_t(j)}{p_t(m)} - \frac{p_t(j)}{p_t(m)}\frac{\exp(\beta_t r(j))}{\exp(\beta_t r(m))}\Big) + \frac{\partial\beta_t}{\partial t}r(i) \tag{169}$$

$$\square$$

# D    EXPERIMENTAL DETAILS

Code is available at `https://anonymous.4open.science/r/discrete_fkc-40B8/`.

## D.1    AMORTIZED LINEAR REGRESSION

### D.1.1    THEORETICAL JUSTIFICATION

The posterior over parameters factors as:

$$p(\theta|\mathcal{X}) \propto p(\theta)p(\mathcal{X}|\theta) = p(\theta) \prod_k^K p(\mathcal{X}_k|\theta) \propto p(\theta)^{1-K} \prod_k^K p(\theta|\mathcal{X}_k) \qquad (170)$$

For a uniform prior $p(\theta)$, this results in the product we applied $p(\theta|\mathcal{X}) \propto \prod_{k=1}^K p(\theta|\mathcal{X}_k)$.

### D.1.2    EXPERIMENTAL SETUP

All experiments were done on a single A100 GPU.

For each experiment, the dataset $\mathcal{X}$ was generated using $(\theta_0, \theta_1) = (3.0, 4.0)$, with $x$ spaced linearly between $[-10, 10]$, and $y_i = \theta_1^* x_i + \theta_0^* + \epsilon$, where $\epsilon \sim \mathcal{N}(0, 0.1^2)$.

For inference with LLaDA, a temperature of $1.0$ was used, and the random remasking strategy was applied. All predictions were made in a single block, and the generation length was capped at $128$ tokens.

The number of SMC samples were selected based on MSE, for a fixed dataset size, over a grid of $\{2,4,5,8,16,32\}$. 5 SMC samples were selected for evaluation.

The prompt used to generate predictions is of the form: `"Assume a model of the form y = a * x + b, where a and b are the parameters of the model. The observations are given as (x,y) points, where y has Gaussian noise with standard deviation 0.1 added.  Predict the parameters of linear regression for (x,y) points:  "` + $(x_1, y_1)$, ..., $(x_N, y_N)$ + `" Output the final answer as:  "The best estimate for parameters of the model are:  a = _, and b = _"` where _ is replaced with the values of a and then b.`"`

The generated data varied for different seeds, meaning that different seeds resulted in different prompts.

### D.1.3    ADDITIONAL RESULTS FOR AMORTIZED LEARNING

We include an ablation over the number of SMC samples, for a fixed number of products in Figure A1. We can observe that more SMC samples improves performance, up to a threshold of $8$ samples.

We additionally include a comparison of how well the outputs adhered to the specified prompt format in Figure A2.

Some selected samples from the product and joint prompting strategies are included in Table A3. We can note that outputs using joint prompting often fail to adhere to the output format specified in the prompt, and sometimes cannot be parsed for values of $(\theta_0, \theta_1)$. This issue wasn't observed for the product prompt (using any number of particles).

## D.2    CODE GENERATION

All experiments were done on an a A100L GPU.

For evaluation, the HumanEval (Chen et al., 2021) and MBPP (Austin et al., 2021) coding datasets were used. For MBPP the sanitized dataset split was evaluated. For MBPP, the prompt was modified to add the function definition.

Evaluation consisted of computing average accuracy on test-cases provided in the dataset. For evaluation, the longest code segment in the generated output without any syntactic errors was parsed and sanitized.

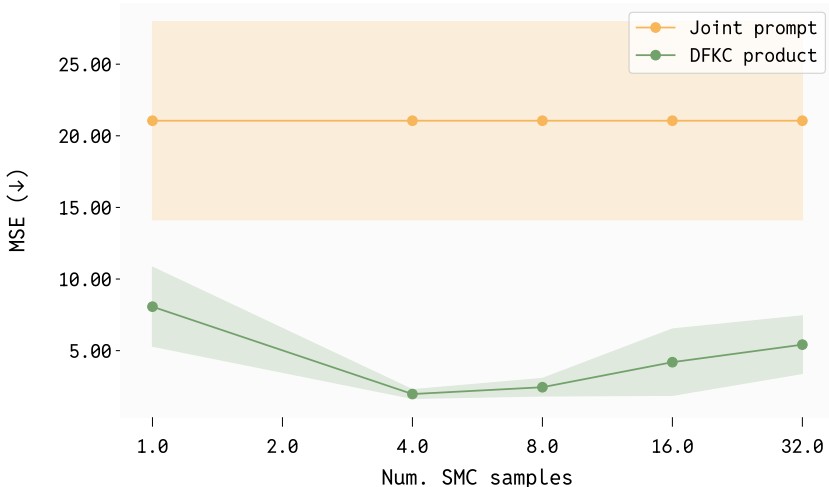

Figure A1: Increasing the number of SMC samples for DFKC improves over no SMC resampling; gain is largest with 4 or 8 samples. Taking the product has a lower (better) mean squared error (MSE) than joint prompting, and resampling with DFKC significantly improves this further.

 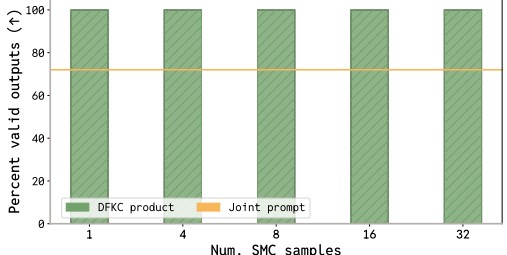

(a) DFKC generates a higher percentage of valid, parseable outputs compared with joint prompting at all data sizes.

(b) DFKC generates consistently generates 100% valid, parseable outputs at all SMC sample sizes while joint prompting only generates 72% valid prompts on average.

Figure A2: Effect of data quantity on predicting linear regression parameters.

Hyperparameters consisted of the number of SMC samples $M$, and inverse-temperature $\beta$. These were selected through search in a grid of $M = \{2, 4, 8\}$ and $\beta = \{3.0, 5.0, 10.0, 20.0\}$. Additionally, a remasking strategy of 'low confidence' or 'random' was evaluated on the hold-out set. The selection was performed based on accuracy on a small validation set (10 prompts from each dataset). For HumanEval, the final results were computed with $M = 4$, $\beta = 10.0$, and random remasking. For MBPP, the final results were computed with $M = 4$, $\beta = 20.0$, and random remasking.

The final evaluation results are reported on the remainder of the dataset (154 datapoints for HumanEval, and 417 points for MBPP).

A generation length of 128 was used for all experiments.

A table of results is included in Table A4 (the same values which were plotted in Figure 4). We note that "Naive Annealing" is equivalent to our method with 1 SMC sample (eg. without resampling). The improvement between our method and Naive Annealing therefore demonstrates the importance of resampling.

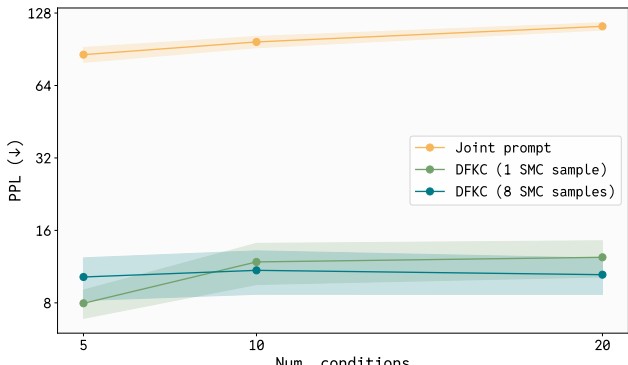

Figure A3: Multi-constraint story generation task: Comparison of Perplexity (PPL), between joint prompting, DFKC (1 SMC sample), and DFKC (8 SMC samples), for different numbers of conditions.

Table A4: Accuracy on coding tasks, with standard error reported over 5 seeds

| Method | Human Eval (%) | MBPP (%) |
|---|---|---|
| Base Model | $12.83 \pm 0.38$ | $10.00 \pm 0.25$ |
| Base Model Argmax | $30.74 \pm 0.76$ | $30.28 \pm 0.87$ |
| Naive Annealing | $30.49 \pm 0.49$ | $29.24 \pm 1.07$ |
| DFKC (Ours) | $\mathbf{33.78 \pm 0.97}$ | $\mathbf{31.00 \pm 0.40}$ |

### D.3 MULTI-CONSTRAINT STORY GENERATION

We evaluate the product formulation for generating stories. For this task we prompt the language model to generate a story, with a list of constraints $C = \cup_k C_k$. Constraints may demand the inclusion of particular events or characters (such as a "hungry cat"), or be stylistic in nature ("the story should have mystery"). We use our method to sample from the product over individual constraints, and evaluate our adherence to the constraints by using the perplexity of the output under a more powerful language model, Qwen2.5 (Yang et al., 2024). Results for our method, over a varying number of constraints $K$, are included in Figure A3.

All experiments were performed on a single L40 GPU.

For inference with LLaDA, the next token to unmask was chosen randomly (as opposed to picking highest confidence one) due to the model frequently sampling end of text tokens using the latter setting. All experiments used a temperature of $T = 1.0$, and was generated within a single block with generation length $L$ varying based on the number of conditions $C$ (to account for the increasing complexity of the task as $C$ grows). In particular:

$$L = \begin{cases} 128, & C \leq 6 \\ 256, & 6 < C \end{cases} \tag{171}$$

The prompt for the story generation is composed as follows: the base prompt is "`Write a story.`". The conditions are sampled at random from a set of 50 conditions, containing mutually compatible constraints such as:

1. "`It should include a curious child.`"
2. "`It should describe a small village.`"
3. "`It should feature a dense forest.`"
4. ...

The number of SMC samples was selected by optimizing for PPL over a grid of $\{4, 8, 12\}$. The final results were computed with 8 samples.

Different seeds resulted in a different set of constraints being sampled to form the prompt.

## D.4 PROTEIN SEQUENCE GENERATION

All experiments were done on a single L40 GPU. For each length and particle type, we generate 50 samples.

The base discrete diffusion model used is DPLM1 650M (Wang et al., 2024b). For a sequence of length $l$, $l$ generation steps are used, and once a token is unmasked, it is not remasked in future steps (to align more closely to the traditional masked diffusion generation process, and as opposed to the remasking strategies used in (Wang et al., 2024b)).

### D.4.1 REWARD MODELS

**ESM2 Likelihood Reward.** The log-reward for a sequence $x$ with length $L$ is defined using the ESM2 model $f_\theta$. We first compute a score $S(x)$ by passing the entire sequence $x$ to the model, and then averaging the log-likelihoods evaluated at the amino acid sequence:

$$S(x) = \frac{1}{L} \sum_{i=1}^{L} f_\theta(x)[x_i] \tag{172}$$

This approach allows us to compute the toy reward in one single pass.

**Thermostability Reward.** We use a fine-tuned version of DPLM-650M which predicts thermostability from sequences. This model was evaluated to have a 0.695 Spearman correlation (see Table 1 of Wang et al. (2024a)). The score $S(x)$ is the log of the predicted thermostability value.

In both cases, the score is scaled by a hyperparameter $\gamma$ to obtain the log-reward $r(x)$:

$$r(x) = \gamma S(x) \tag{173}$$

In our experiments for unconditional protein sequence, we set a hyperparameter $\gamma = 200$ across all lengths and particles for the ESM2 reward, and $\gamma = 10$ for the thermostability reward.

For partially masked sequences $x$, the reward is computed by first denoising $x$ to a completely unmasked sequence $x_0$ by sampling from the denoiser (in a single step), and then evaluating the reward on $x_0$. That is: $r(x) := r(x_0)$, $x_0 \sim p_\theta(x_0|x_t = x)$ (where $p_\theta(x_0|x_t)$ is the denoiser distribution, not the exact posterior).

Evaluating the reward ratio in Theorem 3.5 requires computing the reward on all neighbors of $x$ (sequences that differ from $x$ at a single token position). In this case, two choices are made to make the calculation more tractable:

1. The token position to unmask is chosen prior to computing the reward terms, from the base processes rate matrix (and logits). This means that the reward ratio only needs to be computed on neighbors differing from $x$ on the chosen token position (for a vocabulary of size $V$, and length $L$, this reduces the number of reward evaluations from $LV$ to $V$).

2. In computing the reward $r(x')$ on a partially masked neighbor $x'$: we do not unmask it using a separate call to the denoiser, instead all masked positions in $x'$ are replaced with the corresponding tokens from $x_0 \sim p_\theta(x_0|x_t = x)$, ie. with the denoiser called on the sequence $x$. This is an approximation to computing $x'_0 \sim p(x'_0|x_t = x')$, and avoids multiple calls to the denoiser.

A linear annealing schedule $\beta_t = 1 - t$ is used for the reward (where generation starts at $t = 1$ and proceeds to $t = 0$).

### D.4.2 BASELINES

**DG-Exact.** [(Nisonoff et al., 2024)] This method is equivalent to DFKC without resampling (or equivalently, with 1 SMC sample). Hyperparameters are set the same as our method.

**FKSteering.** [(Singhal et al., 2025)] The FK steering baseline uses the base model as the proposal, and uses the difference potential for resampling. The main hyperparameter involved is the number of

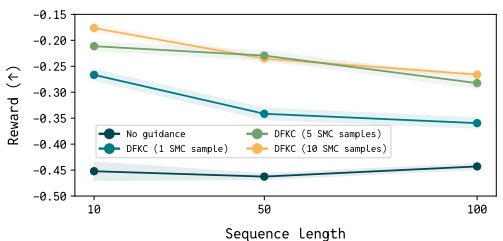 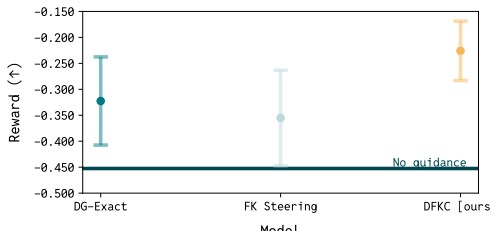

(a) Rewards (ESM2-650M log-likelihood) of generated sequences for 10, 50, 100 amino acids at 1, 5, 10 SMC samples and base model (no guidance).

(b) Comparison of ESM2-650M log-likelihood rewards of best DFKC model with FK Steering (Singhal et al., 2025) and DG-Exact (Nisonoff et al., 2024).

Figure A4: DFKC performance on reward-guided unconditional protein sequence generation.

SMC samples ($M$). The method is evaluated for $M = 5$ and $M = 10$, and the best performance for each reward is selected and reported in Table 2.

### D.4.3 DESCRIPTION OF PROTEIN METRICS

**Diversity.** The "sequence diversity" metric of the generated sequences was obtained by normalizing the global pairwise sequence alignment. The metric serves to quantify how different the sampled sequences are.

Another metric for sequence diversity, "Max cluster", was evaluated using MMseqs2 clustering (Steinegger & Söding, 2017). Generated sequences were clustered with mmseqs easy-cluster using a 50% sequence identity threshold, 80% coverage, and cov-mode 0 (`mmseqs easy-cluster --min-seq-id 0.5 -c 0.8 --cov-mode 0`). Each cluster represents a group of related variants; we measured diversity as the fraction of clusters formed divided by the total number of sequences.

**Structural confidence.** To assess structural plausability and quality, we used ESMFold (Lin et al., 2022) to predict a structure for each sequences and then computed both the average predicted local distance difference test (pLDDT) and predicted TM (pTM) score. pLDDT estimates how confident the model is in the local geometry at each residue, and pTM predicts how correct the overall topology is. High-confidence structures are considered to be those with pLDDT $> 0.7$ and pTM $> 0.7$; we also report this percentage.

**Novelty.** For each sequence, we evaluated each sequence using `BLASTp` against the ClusteredNR protein database from November 20, 2025 (Camacho et al., 2009). If a generated sequence was matched to a sequence in the database, this would be considered as a hit. The less hits, the more novel the generations.

### D.4.4 HYPERPARAMETERS

The main hyperparameters consisted of the reward scale $\gamma$, and the number of SMC samples used in DFKC and FK Steering. The reward scale was selected by evaluating the average (unscaled) reward on a set of generated sequences of smaller length (10, and 20), for $\gamma \in \{1.0, 5.0, 10.0, 50.0, 100.0, 200.0\}$. For the number of SMC samples, we evaluated the relevant methods with 5 and 10 samples, with results written in Table A5. The results for the best performing number of samples for each method was selected and reported in Table 2.

### D.4.5 ADDITIONAL PROTEIN RESULTS

In Figure A4a we plot the protein reward along the evaluated sequence lengths and along different numbers of SMC samples for DFKC. We observe that going to 5 particles from 1 particle (without resampling, which is equivalent to the guidance method referred to as DG-Exact in (Nisonoff et al., 2024)) substantially improves the reward, especially for lengths $> 10$. We note that our method outperforms the base model and DG-Exact at all lengths.

We report the protein metrics for the ESM2 likelihood and Thermostability tasks, varying the number of SMC samples for FKSteering and DFKC, in Table A5.

The highest reward settings for each method was reported in Table 2.

### D.5 ANNEALING THE ISING MODEL

In Figure A5, we compare our method with the ground truth (obtained from long runs of the Swendsen–Wang algorithm with open boundary conditions) and with the theoretical results for an infinite lattice. The comparison is presented in terms of the mean energy and magnetization for various inverse temperatures.

The settings for various experiments are outlined below.

**Dataset for experiment 1**

```
source: synthetic Ising model configurations
size: 100,000 samples after burn-in
sampling method: Swendsen-Wang
    burn-in length: 10,000 steps
    thinning interval: 5
beta: 0.25
lattice size: 16
```

**Dataset for experiment 2**

```
source: synthetic Ising model configurations
size: 10,000 samples after burn-in
sampling method: Glauber dynamics
    burn-in length: 10,000 steps
    thinning interval: 1
beta: 0.2 and 0.3
lattice size: 16
```

**Model**

```
architecture: UNet
activation: SiLU
channels: [16, 32, 64]
resblocks per stage: 2
attention: applied at 4×4 resolution
initialization: Xavier uniform
time embedding: sinusoidal embedding
```

**Training**

```
optimizer: AdamW
    learning rate: 2e-4
    betas: (0.9, 0.999)
    weight_decay: 1e-4
batch size: 400
epochs: 6000
learning rate schedule: constant with warmup
hardware: 1 × NVIDIA A100 GPU (40 GB memory)
loss: denoising score entropy
```

**Evaluation**

```
metrics for global structure: 2-Wasserstein metric between
distributions of
energy and distributions of magnetization.
metrics for local structure: MSE for correlation function.
sample size: 10,000
```

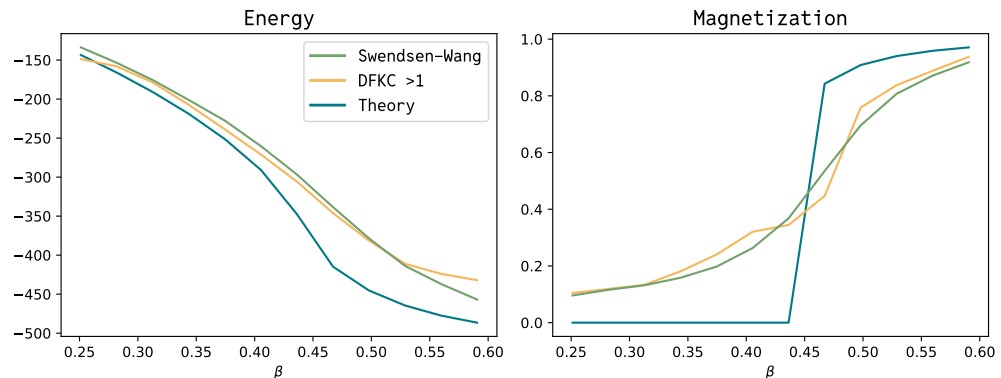

Figure A5: Average energy and magnetization for DFKC, Swendsen-Wang algorithm with open boundary conditions and theoretical. Training $\beta$ for DFKC is 0.25.

The hyperparameters searched over are outlined below.

```
optimizer:
    learning rate: [5e-5, 1e-4, 2e-4, 3e-4, 4e-4, 1e-3]
    weight_decay: [0, 5e-5, 1e-4, 5e-4, 1e-3]
batch size: [100, 200, 400, 800, 1600]
model:
    UNet base channels: [16,32,64,128]
    time embedding size: [16,32,64,128]
rate matrix time dependence: [linear, sine]
method:
    Number of particles: [100, 1000, 5000, 10000]
    Number of steps: [500, 1000, 2000, 5000]
```

Hyperparameters were selected based on Wasserstein-2 distance between samples and Swendsen-Wang distance for energy distribution. For the DFKC method itself we looked at number of steps and number of particles as hyperparameters.

| Data size | Joint Prompt Output | Product Prompt Output |
|---|---|---|
| N=10 | The best estimate for parameters of the model are: a = 4.337, and b = -34.049 | The best estimate for parameters of the model are: a = 3.000, and b = 10.004 |
| N=20 | Based on the observed data points, we can see a trend that y is directly proportional to x. The best estimate for the parameters a and b is: a = 1.0, and b = 0.0. | The best estimate for parameters of the model are: a = 3.82, and b = 10.12. |
| N=50 | To obtain the best estimates for the parameters (a and b), you need to follow the detailed steps of building a linear regression model using Ordinary Least Squares (also namedIM, and guide, filter). These steps involve typically a program such as R or a statistical tool among others. The objective is to predict parameters, but after an ensemble calculation, we are going to use, known as the sum of residuals, to estimate the model's parameters. The sum of residuals helps us evaluate the discrepancy of model with a given residuals. Once I've made these predictions, I'll be able to provide more precise feedback on parameter estimates. | The best estimate for parameters of the model are: a = 1.344, and b = -22.331 |
| N=100 | The best estimate for parameters of the model are: a = 0x583C622F 052D29A9 + 00EA6F242949D26F and b = 0x 41796E30 0027A200 - 76CF406498D45505. Note: These values of a and b are with 95% confidence taking into account the Gaussian balls added to Python and Python recovery points. | The best estimate for parameters of the model are: a = 0.8313, and b = 0.0564. |

Table A3: Comparison between curated joint and product prompt outputs at varying data sizes.

Table A5: Ablation over Number of SMC Samples ($M$) for Protein Tasks, (over 5 seeds)

| | Reward | Diversity | | Structural confidence | | | Novelty |
|---|---|---|---|---|---|---|---|
| | $\log r(x)$ ($\uparrow$) | Seq. diversity ($\uparrow$) | Max. cluster ($\uparrow$) | pLDDT ($\uparrow$) | pTM ($\uparrow$) | Frac. (pLDDT & pTM) > 0.7 ($\uparrow$) | Frac. BLASTp hits ($\downarrow$) |
| **Task: unconditional generation** | | | | | | | |
| Base [unguided] | $-0.4520 \pm 0.0676$ | $\mathbf{0.7729 \pm 0.0570}$ | $\mathbf{0.3333}$ | $0.5827 \pm 0.1539$ | $0.2535 \pm 0.1403$ | $0.00$ | $\mathbf{0.0267}$ |
| FK Steering ($M = 5$) Singhal et al. (2025) | $-0.3553 \pm 0.0929$ | $0.6978 \pm 0.2167$ | $0.1067$ | $0.5765 \pm 0.1686$ | $0.2098 \pm 0.1168$ | $0.00$ | $0.0333$ |
| FK Steering ($M = 10$) Singhal et al. (2025) | $-0.3654 \pm 0.0658$ | $0.6226 \pm 0.2923$ | $0.0733$ | $0.6381 \pm 0.1859$ | $0.3570 \pm 0.2400$ | $0.15$ | $0.1333$ |
| DFKC ($M = 5$) [ours] | $-0.2411 \pm 0.0666$ | $0.6815 \pm 0.2211$ | $0.0867$ | $0.6398 \pm 0.1450$ | $0.3144 \pm 0.2035$ | $0.05$ | $0.1667$ |
| DFKC ($M = 10$) [ours] | $\mathbf{-0.2259 \pm 0.0415}$ | $0.6144 \pm 0.2954$ | $0.0533$ | $\mathbf{0.7148 \pm 0.1416}$ | $\mathbf{0.5006 \pm 0.2120}$ | $\mathbf{0.19}$ | $0.4133$ |
| **Task: thermostability** | | | | | | | |
| Base [unguided] | $-0.6590 \pm 0.1026$ | $\mathbf{0.7723 \pm 0.0582}$ | $\mathbf{0.3571}$ | $\mathbf{0.5941 \pm 0.1525}$ | $\mathbf{0.2609 \pm 0.1452}$ | $0.00$ | $0.0286$ |
| FK Steering ($M = 5$) Singhal et al. (2025) | $-0.5841 \pm 0.1360$ | $0.7513 \pm 0.1723$ | $0.2733$ | $0.5704 \pm 0.1534$ | $0.2246 \pm 0.1087$ | $0.00$ | $0.0400$ |
| FK Steering ($M = 10$) Singhal et al. (2025) | $-0.5980 \pm 0.0956$ | $0.6944 \pm 0.2197$ | $0.2467$ | $0.5473 \pm 0.1495$ | $0.1901 \pm 0.0718$ | $0.00$ | $\mathbf{0.0133}$ |
| DFKC ($M = 5$) [ours] | $\mathbf{-0.5316 \pm 0.1082}$ | $0.7618 \pm 0.1302$ | $0.3200$ | $0.5875 \pm 0.1517$ | $0.2468 \pm 0.1387$ | $0.01$ | $0.0533$ |
| DFKC ($M = 10$) [ours] | $-0.5736 \pm 0.0974$ | $0.7554 \pm 0.1503$ | $0.3200$ | $0.5866 \pm 0.1451$ | $0.2501 \pm 0.1508$ | $\mathbf{0.03}$ | $0.0533$ |

