# OpenReview forum: "Discrete Feynman-Kac Correctors"
_ICLR.cc/2026/Conference — Submitted to ICLR 2026_

### Official Review · Reviewer_AQs3 · 2025-10-22

**Soundness:** 3
**Presentation:** 3
**Contribution:** 2
**Rating:** 4
**Confidence:** 3

**Summary:**

The article under consideration considers the problem of correcting the output of a pre-trained generative modeling. More specifically, assuming that $ p_0 $ is the output of the pre-trained model, the following tasks are considered

- *Annealing* One wants to sample from the law propositional to to $p_0^{\beta}$ where $\beta$
- *Product and geometric averaging* Here, one wants to sample from the product of two pretained models $p^1_0p^2_0$
- *Reward-tilting* One wants to sample from the law propositional  to $p_0\exp(r)$ where $r$ is a reward function.

For all these tasks, the authors derive a Fokker-Planck Equation (FPE) for the modified flows given by $p^\beta_t$,$p^1_tp^2_t,p_t\exp(r)$ up to normalizing constants. This FPE involves modified jump rates $B_t(i,j)$, whose expression is given in terms of the original rates and the probability rations $p_t(i)/p_t(j)$, and a non linear Feynman-Kac term $g_t$. The authors exploit the structure of this equation to apply  Sequential Monte Carlo algorithms (SMC) to sample from the target modified distribution. The paper is completed by numerical experiments, one per task considered: temperature annealing for the Ising model, text generation and protein guidance for reward tilting.

**Strengths:**

The paper proposes a seemingly new method to tackle the general problem of modifying the output of a pre-trained model. Remarkably, this method does not require any extra training step or fine-tuning. Numerical experiments are conducted across a quite large range of tasks and show promising results.

**Weaknesses:**

- From a mathematical standpoint, the main weakness is that there is no theoretical guarantee of convergence for the proposed methodology. I would like very much to see som e result in this direction. The derivation of the FPE for the modified flow is interesting, but follows from a rather standard calculation. This does not mean that is unimportant, of course.

- From the methodological perspective, it appears to me that the main message is to use SMCs instead of other methods for reward-tilting and other tasks. The final methodology is then obtained assembling together two class of algorithms: SMCs and generative modeling, but I do not see new algorithmical ideas emerging from the paper. I may be wrong and I would be happy to review my assessment if the authors bring convincing evidence about the novelty of their methodology. I am not sure the experiments alone are strong enough to warrant publication. I am not the best person to assess their validity and I will abstain from judging them in detail.

**Questions:**

- I would expect to see the value of the parameters in the modified flows to vary over time. For example, I would expect to see the annealing temperature in Thm 3.1 to depend on $t$ so that $\beta_1=0$ and $\beta_0=\beta$, where . Is there a reason for not implementing this in practice?

---

> ### Author Response · Authors · 2025-11-24
> **Author Response to Reviewer AQs3 (part 1)**
>
> We thank the reviewer for their time in reviewing our work and providing constructive comments. We were glad to see that the reviewer found our results promising across a “large range of tasks”. Below, we address each of the points raised in the review:
>
> > From a mathematical standpoint, the main weakness is that there is no theoretical guarantee of convergence for the proposed methodology. I would like very much to see some result in this direction. The derivation of the FPE for the modified flow is interesting, but follows from a rather standard calculation. This does not mean that is unimportant, of course.
> >
>
> Thank you for acknowledging the importance of the proposed results! Indeed, a deeper theoretical analysis of Feynman-Kac correctors is of a great interest. For now, we provide the proof of the Feynman-Kac formula (**Theorem 2.1 in the revised paper**), which establishes the convergence of the time-discretization scheme. Establishing the convergence *bounds* on the sampling scheme requires development of more advanced tools akin to those proposed in [1,2]. These, however, would require an extensive theoretical investigation, meriting an independent study and publication. **We add a discussion regarding theoretical guarantees to the related works section in the revised manuscript**.
>
> [1] Ren, Yinuo, Haoxuan Chen, Grant M. Rotskoff, and Lexing Ying. "How discrete and continuous diffusion meet: Comprehensive analysis of discrete diffusion models via a stochastic integral framework." *arXiv preprint arXiv:2410.03601* (2024).
>
> [2] Benton, Joe, Valentin De Bortoli, Arnaud Doucet, and George Deligiannidis. "Nearly $ d $-linear convergence bounds for diffusion models via stochastic localization." *arXiv preprint arXiv:2308.03686* (2023).
>
> > From the methodological perspective, it appears to me that the main message is to use SMCs instead of other methods for reward-tilting and other tasks. The final methodology is then obtained assembling together two class of algorithms: SMCs and generative modeling, but I do not see new algorithmical ideas emerging from the paper. I may be wrong and I would be happy to review my assessment if the authors bring convincing evidence about the novelty of their methodology.
> >
>
> The main methodological novelty is leveraging the available information in the model (such as the ratio of marginal probabilities) to develop an SMC method which, as we demonstrate, unlocks a principled and efficient way of controlling discrete diffusion models. In particular, our algorithm allows for sampling from the annealed or product distributions of these models, despite these densities being intractable to evaluate. Additionally, our framework provides guarantees on the generated marginals (see Theorem 2.1), which is not the case for re-weighting proportionally to a reward function [3].  Moreover, DFKC does not require running Markov Chain Monte Carlo methods [4,5] or any fine-tuning [6], which would significantly increase the computational budget required for the algorithm.
>
> [3] Singhal, Raghav, Zachary Horvitz, Ryan Teehan, Mengye Ren, Zhou Yu, Kathleen McKeown, and Rajesh Ranganath. "A general framework for inference-time scaling and steering of diffusion models." *arXiv preprint arXiv:2501.06848* (2025).
>
> [4] Du, Yilun, Conor Durkan, Robin Strudel, Joshua B. Tenenbaum, Sander Dieleman, Rob Fergus, Jascha Sohl-Dickstein, Arnaud Doucet, and Will Sussman Grathwohl. "Reduce, reuse, recycle: Compositional generation with energy-based diffusion models and mcmc." In *International conference on machine learning*, pp. 8489-8510. PMLR, 2023.
>
> [5] Karan, Aayush, and Yilun Du. "Reasoning with sampling: Your base model is smarter than you think." *arXiv preprint arXiv:2510.14901* (2025).
>
> [6] Domingo-Enrich, Carles, Michal Drozdzal, Brian Karrer, and Ricky TQ Chen. "Adjoint matching: Fine-tuning flow and diffusion generative models with memoryless stochastic optimal control." *arXiv preprint arXiv:2409.08861* (2024).
>
> > I am not sure the experiments alone are strong enough to warrant publication. I am not the best person to assess their validity and I will abstain from judging them in detail.
> >
>
> We have extended the empirical study to demonstrate the utility of DFKC on a diverse set of tasks. Notably, DFKC allows for substantial improvements for the code generation task (**included in Figure 4 of the revised paper**) as well as for the design of proteins with high thermostability (**included in Table 2 of the revised paper**).

---

> > ### Comment · Reviewer_AQs3 · 2025-11-25
> >
> > I thank the authors for addressing my questions. I record that developing convergence guarantees is out of the scope of this work.
> >
> >
> > I don't understand why Theorem 2.1 "establishes the convergence of the time-discretization scheme". To me this is just Feynman-Kac formula for a continuous time Markov chain. I see some time-discretization argument in the proof (which can be avoided using Itô formula, I guess) but none in the statement of the theorem. Am I missing something here?

---

> > > ### Author Response · Authors · 2025-11-26
> > >
> > > We thank the reviewer for their prompt response. By the “convergence of the time-discretization scheme” we meant that the expectations taken over $X_{0:T}$ of Theorem 2.1 were defined through limits of the discrete-time transition kernels (Equation 3).
> > >
> > > Namely if we define the discrete-time estimate $a_{T, dt}$:
> > >
> > > $$
> > > \begin{align}
> > > a_{T, dt} &= \sum_{x_{T}}\ldots \sum_{x_{0}}\phi(x_{T})p(x_{T}\mid x_{T-dt})\ldots p(x_{dt} \mid x_{0})\exp\left(\sum_{t=0}^{T} dt \bar{g}{t}(x{t})\right)p_0(x_{0})\nonumber
> > > \end{align}
> > > $$
> > >
> > > Then Theorem 2.1 establishes that $E_{p_T(x)}[\phi(x)] = \lim_{dt \to 0} a_{T,dt}$
> > >
> > > We have written this more explicitly in the statement of Theorem 2.1 in the revised manuscript.
> > >
> > > We welcome any suggestions to convey these results more clearly.

---

> ### Author Response · Authors · 2025-11-24
> **Author Response to Reviewer AQs3 (part 2)**
>
> ***
> **Questions:**
>
> > I would expect to see the value of the parameters in the modified flows to vary over time. For example, I would expect to see the annealing temperature in Thm 3.1 to depend on $t$ so that $\beta_1 = 0$ and $\beta_0 = \beta$, where . Is there a reason for not implementing this in practice?
> >
>
> Indeed, DFKC accommodates a time-dependent temperature $\beta_t$ in the annealing algorithm in the same way it handles a time-dependent reward in Theorem 3.5. However, this would add an additional time-derivative term to the weight update in equation (13)
>
> $$
> \begin{align}
> g_t(i) &= \ldots + \frac{\partial \beta_t}{\partial t} \log q_t(i)
> \end{align}
> $$
>
> Although, the time-derivative of $\beta_t$ is available, this term requires the evaluation of the marginal density $\log q_t(i)$, which is not immediately available in practice (unlike the ratio of marginals). Evaluation of the density can be done either by training an energy-based model (as proposed in [7]) or using a density estimator (e.g., from [8,9]). However, both approaches introduce additional difficulties without obvious benefits; hence, we leave this degree of freedom for future studies.
>
> [7] Akhound-Sadegh, Tara, Jungyoon Lee, Avishek Joey Bose, Valentin De Bortoli, Arnaud Doucet, Michael M. Bronstein, Dominique Beaini, Siamak Ravanbakhsh, Kirill Neklyudov, and Alexander Tong. "Progressive Inference-Time Annealing of Diffusion Models for Sampling from Boltzmann Densities." *arXiv preprint arXiv:2506.16471* (2025).
>
> [8] Skreta, Marta, Lazar Atanackovic, Avishek Joey Bose, Alexander Tong, and Kirill Neklyudov. "The Superposition of Diffusion Models Using the It\^ o Density Estimator." *arXiv preprint arXiv:2412.17762* (2024).
>
> [9] Karczewski, Rafał, Markus Heinonen, and Vikas Garg. "Diffusion Models as Cartoonists: The Curious Case of High Density Regions." *arXiv preprint arXiv:2411.01293* (2024).

---

> > ### Comment · Reviewer_AQs3 · 2025-11-25
> >
> > Thank you for clarifying this. Indeed, the evaluation of the marginal density is challenging.

---

### Official Review · Reviewer_wKyB · 2025-10-23

**Soundness:** 4
**Presentation:** 3
**Contribution:** 3
**Rating:** 8
**Confidence:** 3

**Summary:**

This paper introduces Discrete Feynman-Kac Correctors, a framework that enables flexible control over the generated distribution of pretrained discrete masked diffusion models at inference time. Using Sequential Monte Carlo (SMC) algorithms, the method allows for temperature annealing, sampling from products of marginals, and integrating external reward functions—all without additional training. The approach is demonstrated on the Ising model, language modeling, multi-constrained generation, and protein sequence generation, offering a versatile tool for controlled sampling.

**Strengths:**

- The paper presents a theoretically sound and novel approach.
- The method is rigorously evaluated across a broad spectrum of benchmarks, demonstrating its versatility and potential applicability.

**Weaknesses:**

- The benchmarks on the Ising model lack comparison to theoretically available ground truth solutions, which would strengthen the validation of the proposed method.
- The evaluation does not include the critical temperature regime of the Ising model, where sampling is known to be particularly challenging.
- The notation $g_t(i)$ appears in Line 122 but is not formally introduced or defined elsewhere in the text.
- With the exception of Figure 4b, the paper does not provide direct comparisons to alternative methods, limiting the ability to contextualize the performance of the proposed approach.

**Questions:**

- How does the method perform on the Ising model at the critical temperature? Would it be possible for the authors to compare their results to theoretically derived values for the Ising model with periodic boundary conditions, as outlined in [1]? The following publicly available script could be used to compute these values for comparison:
 [link to script](https://github.com/ml-jku/DiffUCO/blob/main/IsingTheoryBaselines/IsingTheory.py)

**Minor Comment:**
- The related work section could be enriched by discussing recent advances in discrete diffusion samplers [2] and discrete flow samplers [3, 4]. These works also focus on sampling from unnormalized target distributions, albeit through learned rather than guided approaches, and include evaluations on the Ising model.

**References:**
[1] Arthur E. Ferdinand and Michael E. Fisher. "Bounded and inhomogeneous Ising models. I. Specific-heat anomaly of a finite lattice." *Physical Review*, 185(2):832, 1969.

[2] Sanokowski, Sebastian, et al. "Scalable Discrete Diffusion Samplers: Combinatorial Optimization and Statistical Physics."

[3] Holderrieth, Peter, Michael Samuel Albergo, and Tommi Jaakkola. "LEAPS: A discrete neural sampler via locally equivariant networks." *Forty-second International Conference on Machine Learning*.

[4] Ou, Zijing, Ruixiang Zhang, and Yingzhen Li. "Discrete Neural Flow Samplers with Locally Equivariant Transformer." *arXiv preprint arXiv:2505.17741* (2025).

---

> ### Author Response · Authors · 2025-11-24
> **Author Response to Reviewer wKyB (part 1)**
>
> We thank the reviewer for the very concrete and detailed suggestions for improving our manuscript. We are delighted to learn that they found our approach “theoretically sound and novel”, and that they judged our method to be evaluated on “a broad spectrum of benchmarks, demonstrating its versatility and potential applicability”.
>
> We address and incorporate the suggestions proposed by the reviewer below.
>
> > The benchmarks on the Ising model lack comparison to theoretically available ground truth solutions, which would strengthen the validation of the proposed method.
> >
>
> We thank the reviewer for this suggestion. We have added a comparison with theoretical predictions as well as with samples generated by the Swendsen-Wang algorithm **in Figure A5 of Appendix D.5**. In the current version, our model is trained on Swendsen-Wang samples for the Ising model with open boundary conditions. As a result, our estimates of energy and magnetization differ slightly from the theoretical values due to finite-size and boundary condition effects.
>
> > The evaluation does not include the critical temperature regime of the Ising model, where sampling is known to be particularly challenging.
> >
>
> We agree that investigating the critical temperature regime of the Ising model is both particularly interesting and challenging. In the revised submission, we extend our evaluation to include and go beyond the critical point, with results included **in Figure 2**. In particular, we demonstrate that our method, starting from a model trained at $\beta = 0.25$, is able to sample in the range $\beta \in[0.25, 0.6]$ (the critical temperature is $\beta_c \approx 0.4407$) with an accuracy matching or exceeding the baselines.
>
> > The notation g_t(i) appears in Line 122 but is not formally introduced or defined elsewhere in the text.
> >
>
> We thank the reviewer for pointing this out. The intention was that $g_t(i)$ was an arbitrary function that performs reweighting (to be instantiated later on with specific formulas depending on the target marginals). In the revised manuscript, we have phrased this more explicitly.
>
> > With the exception of Figure 4b, the paper does not provide direct comparisons to alternative methods, limiting the ability to contextualize the performance of the proposed approach.
> >
>
> We have expanded our empirical evaluations, with additional tasks and baselines.
>
> - For sampling from the Ising model, we have added a comparison with the LEAPS method, trained to sample with an annealing schedule up to the critical temperature [1]. These results are included **in Figure 2 of the revised paper**. We observe that our method samples more accurately past the training range of the LEAPS model.
> - For language modelling, we added new experiments for code generation: evaluating the accuracy (the percentage of test cases) passed by the language model for various programming problems. **These results are included in Figure 4 of the revised paper,** and produced below for convenience. For this setting, we compared our method to using alternative sampling strategies with the base model (such as “argmax sampling” - selecting the most likely token at each step). Our method outperforms the investigated alternatives.
> |  | Human Eval Accuracy (%)$(\uparrow)$ | MBPP Accuracy (%) $(\uparrow)$ |
> | --- | --- | --- |
> | Base Model | $12.83  \pm 0.38$ | $10.00 \pm 0.25$ |
> | Base Model (Argmax sampling) | $30.74 \pm 0.76$  | $30.28 \pm 0.87$ |
> | Naive Annealing  | $30.49 \pm 0.49$ | $29.24 \pm 1.07$ |
> | **DFKC (Ours)** | $33.78 \pm 0.97$  | $31.00 \pm 0.40$  |
>
> [1] Peter Holderrieth, Michael Samuel Albergo, and Tommi Jaakkola. LEAPS: A discrete neural sampler
> via locally equivariant networks. International Conference on Machine Learning (ICML), 2025.
> URL https://openreview.net/forum?id=Hq2RniQAET.

---

> ### Author Response · Authors · 2025-11-24
> **Author Response to Reviewer wKyB (part 2)**
>
> > How does the method perform on the Ising model at the critical temperature? Would it be possible for the authors to compare their results to theoretically derived values for the Ising model with periodic boundary conditions, as outlined in [1]? The following publicly available script could be used to compute these values for comparison: [link to script](https://github.com/ml-jku/DiffUCO/blob/main/IsingTheoryBaselines/IsingTheory.py)
> >
>
> We thank the reviewer for raising these points. We address them in detail in our responses to Weaknesses 1 and 2 above. Briefly, we have now (i) extended our experiments to include and go beyond the critical temperature (**which is reported in Figure 2 of the revised paper**) and (ii) added comparisons to theoretical predictions and Swendsen–Wang samples (**Figure A5 in Appendix D.5**).
>
> > The related work section could be enriched by discussing recent advances in discrete diffusion samplers [2] and discrete flow samplers [3, 4]. These works also focus on sampling from unnormalized target distributions, albeit through learned rather than guided approaches, and include evaluations on the Ising model.
> >
>
> We thank the reviewer for this suggestion. We have added a discussion of recent work on neural samplers and diffusion-based approaches [1, 2, 3] to the **Related Work section.**
>
> [2] Sanokowski, Sebastian, et al. "Scalable Discrete Diffusion Samplers: Combinatorial Optimization and Statistical Physics."
>
> [3] Ou, Zijing, Ruixiang Zhang, and Yingzhen Li. "Discrete Neural Flow Samplers with Locally Equivariant Transformer." *arXiv preprint arXiv:2505.17741* (2025).

---

> > ### Comment · Reviewer_wKyB · 2025-11-26
> > **Response to rebuttal**
> >
> > The authors have addressed most of my questions satisfactorily. As I initially assigned a high score of 8, I will maintain this rating. I strongly recommend this paper for acceptance.

---

### Official Review · Reviewer_yXoH · 2025-10-31

**Soundness:** 2
**Presentation:** 3
**Contribution:** 3
**Rating:** 4
**Confidence:** 4

**Summary:**

The paper introduces FKC to discrete diffusion models which are based on jump processes. This framework enables inference alignment of discrete diffusion models without retraining. The key contribution is the derivation of theoretical results showing how annealing, distribution product formation, and reward tilting can be achieved by reweighting and SMC methods. Empirical results demonstrate the applicability of DFKC across three domains: sampling from the Ising model, language modeling, and protein sequence generation.

**Strengths:**

- The manuscript is well-organized and clearly written. The exposition is concise yet thorough, facilitating a clear understanding of the core contributions and methodologies.
 - The paper presents theoretically rigorous and well-founded derivations. The mathematical treatment of annealing, distribution product formation, and reward tilting via reweighting and SMC methods is sound.

**Weaknesses:**

- Unclear motivation: Given the presented and evaluated inference alignment strategies, their usefulness is not convincingly demonstrated. While there is previous work showing that annealing can be beneficial when sampling from Boltzmann distributions, the current manuscript presents this possible advantage for using FKC only in a rather toy-like experiment. Potential benefits for language models are only shown for synthetic and toy tasks rather than real world language modeling at scale. Reward guidance for protein sequence generation is not thouroughly evaluated with additional metrics on quality, diversity, distributional similarity.
 - Limited novelty: FKC is adapted from diffusion to jump processes, the essence of FKC was already presented in Skreta et al. (2025). The re-weighting approach is immediate when transitioning from diffusion to jump processes. From this CTMC formulation the presented proofs are straight-forward derivations and rather mechanical.
 - Limited experimental depth: Despite the appealing breadth of evaluated domains, the depth and rigor of the experimental analysis are limited. Sampling from Boltzmann distributions should have been done for more challenging tasks such as Maximum Independent Set, Maximum Cut as done in previous work. "Armortized Learning" is a synthetic task and "Multi-constraint Story Generation" is a toy task that does not convincingly show real-world benefits in larger-scale language modeling tasks. The protein design experiments lack diverse metrics and thourough comparison with reward guidance based baselines. For all domains, a thorough comparison/discussion and cost/benefit tradeoff analysis of FKC (in particular SMC) compared to inference alignment baselines (Singhal et al., 2025; Nisonoff et al., 2024), reward fine-tuning (Rector-Brooks et al., 2024), and standard diffusion models (more diffusion/euler-maruyama steps, more capacity, longer training) is missing.
 - Datasplits and hyperparameter selection procedures for all compared methods are not described in detail.

**Questions:**

- What are the details on datasplits and hyperparameter selection procedures for all compared methods?

---

> ### Author Response · Authors · 2025-11-24
> **Author Response to Reviewer yXoH (part 1)**
>
> We thank the reviewer for their detailed and constructive feedback. We appreciate that the reviewer found our work well-written and theoretically rigorous. We discuss each concern and question below:
>
> > Unclear motivation: Given the presented and evaluated inference alignment strategies, their usefulness is not convincingly demonstrated. While there is previous work showing that annealing can be beneficial when sampling from Boltzmann distributions, the current manuscript presents this possible advantage for using FKC only in a rather toy-like experiment. Potential benefits for language models are only shown for synthetic and toy tasks rather than real world language modeling at scale.  Reward guidance for protein sequence generation is not thoroughly evaluated with additional metrics on quality, diversity, distributional similarity.
> >
>
> and
>
> > Limited experimental depth: Despite the appealing breadth of evaluated domains, the depth and rigor of the experimental analysis are limited. Sampling from Boltzmann distributions should have been done for more challenging tasks such as Maximum Independent Set, Maximum Cut as done in previous work. "Amortized Learning" is a synthetic task and "Multi-constraint Story Generation" is a toy task that does not convincingly show real-world benefits in larger-scale language modeling tasks. The protein design experiments lack diverse metrics and thorough comparison with reward guidance based baselines.
> >
>
> We address these comments by adding several new experiments and evaluations:
>
> 1. **Boltzmann Sampling**: Based on the reviewer’s comments, we have strengthened our empirical evaluations concerning Boltzmann sampling by extending the Ising model experiments to more challenging regimes and adding a strong diffusion baseline.
>
>     Concretely, we now demonstrate that our method can reliably sample beyond the critical temperature using only samples from the high-temperature regime. In the original submission, our plots covered the interval $\beta \in [0.25, 0.4]$: we drew samples at $\beta=0.25$ with the Swendsen–Wang algorithm and evaluated our method up to $\beta=0.4$, while the critical point is $\beta_c\approx0.4407$. In the revision, we extend this evaluation to $\beta=0.6$ and show that we obtain sample quality comparable to methods that have access to energy evaluations at each $\beta$. We also introduce a new baseline, LEAPS [1], a discrete diffusion sampler trained to sample in the interval $\beta\in[0,\beta_c]$. Our results indicate that our approach matches LEAPS below the critical point while additionally retaining the ability to sample at higher $\beta$, where LEAPS cannot operate effectively due to its training range. **We include these results in Figure 2 of the revised paper.**
>
>
> [1]  Peter Holderrieth, Michael Samuel Albergo, and Tommi Jaakkola. LEAPS: A discrete neural sampler
> via locally equivariant networks. International Conference on Machine Learning (ICML), 2025.
> URL https://openreview.net/forum?id=Hq2RniQAET.

---

> ### Author Response · Authors · 2025-11-24
> **Author Response to Reviewer yXoH (part 2)**
>
> 2. **Language modelling:** Following the reviewer’s suggestion, we add experiments for a code generation task, which presents a more challenging real world application. In particular, we take inspiration from works such as [2, 3] which argue that sampling from higher likelihood regions of an existing language model can lead to better task performance. We evaluate our temperature annealing framework on the task of solving programming problems, over two coding datasets, HumanEval and MBPP [4, 5].  We demonstrate that annealing to a lower temperature using our method improves the accuracy of the LLaDA-8B-Instruct model over alternative sampling methods, with results reported in the table below. **We also revise the main paper and include these results in Figure 4.** Accuracies (percentage of test cases passed) and standard error are reported over 5 seeds
> |  | Human Eval Accuracy (%)$(\uparrow)$ | MBPP Accuracy (%) $(\uparrow)$ |
> | --- | --- | --- |
> | Base Model | $12.83  \pm 0.38$ | $10.00 \pm 0.25$ |
> | Base Model (Argmax sampling) | $30.74 \pm 0.76$  | $30.28 \pm 0.87$ |
> | Naive Annealing  | $30.49 \pm 0.49$ | $29.24 \pm 1.07$ |
> | **DFKC (Ours)** | $33.78 \pm 0.97$  | $31.00 \pm 0.40$  |
>
> 3. **Extended Evaluations for Protein Generation:** As requested by the reviewer, we report several additional metrics for the protein generation task. Furthermore, to demonstrate the universality of the proposed approach we add experiments with a new reward function that measures the thermostability of the protein sequence. **These results are reported in the Table 2 of the revised paper.**
>
>     In particular, the added metrics evaluate:
>
>     - **Diversity**: we compute the global pairwise sequence alignment as well as the number of clusters generated using MMSeqs [6].
>     - **Structural Confidence**: we evaluate the plDDT and pTM scores, computed using ESMFold.
>     - **Novelty**: we evaluate the fraction of sequences which were found to have similarity in an existing protein database (using BLASTp).
>
>     Additional details on protein metrics are included in Appendix D.4.3.
>
>     The results concerning the new thermostability reward (also reported in Table 2) show that our method attains the highest thermostability, while maintaining performance on other metrics comparable to the base model. These results reinforce the efficacy of our method.
>
>
> [2] Audrey Huang, Adam Block, Dylan J. Foster, Dhruv Rohatgi, Cyril Zhang, Max Simchowitz,
> Jordan T. Ash, and Akshay Krishnamurthy. Self-improvement in language models: The sharpening
> mechanism, 2024. URL https://arxiv.org/abs/2412.01951.
>
> [3] Aayush Karan and Yilun Du. Reasoning with sampling: Your base model is smarter than you think,
> 2025. URL https://arxiv.org/abs/2510.14901.
>
> [4] Mark Chen, Jerry Tworek, Heewoo Jun, Qiming Yuan, Henrique Ponde de Oliveira Pinto, Jared
> Kaplan, Harri Edwards, Yuri Burda, Nicholas Joseph, Greg Brockman, Alex Ray, Raul Puri,
> Gretchen Krueger, Michael Petrov, Heidy Khlaaf, Girish Sastry, Pamela Mishkin, Brooke Chan,
> Scott Gray, Nick Ryder, Mikhail Pavlov, Alethea Power, Lukasz Kaiser, Mohammad Bavarian,
> Clemens Winter, Philippe Tillet, Felipe Petroski Such, Dave Cummings, Matthias Plappert, Fotios
> Chantzis, Elizabeth Barnes, Ariel Herbert-Voss, William Hebgen Guss, Alex Nichol, Alex Paino,
> Nikolas Tezak, Jie Tang, Igor Babuschkin, Suchir Balaji, Shantanu Jain, William Saunders,
> Christopher Hesse, Andrew N. Carr, Jan Leike, Josh Achiam, Vedant Misra, Evan Morikawa,
> Alec Radford, Matthew Knight, Miles Brundage, Mira Murati, Katie Mayer, Peter Welinder, Bob
> McGrew, Dario Amodei, Sam McCandlish, Ilya Sutskever, and Wojciech Zaremba. Evaluating
> large language models trained on code, 2021.
>
> [5] Jacob Austin, Augustus Odena, Maxwell Nye, Maarten Bosma, Henryk Michalewski, David Dohan,
> Ellen Jiang, Carrie Cai, Michael Terry, Quoc Le, and Charles Sutton. Program synthesis with large
> language models, 2021. URL https://arxiv.org/abs/2108.07732.
>
> [6] Martin Steinegger and Johannes Söding. Mmseqs2 enables sensitive protein sequence searching for the analysis of massive data sets. Nature biotechnology, 35(11):1026–1028, 2017.

---

> ### Author Response · Authors · 2025-11-24
> **Author Response to Reviewer yXoH (part 3)**
>
> > For all domains, a thorough comparison/discussion and cost/benefit tradeoff analysis of FKC (in particular SMC) compared to inference alignment baselines (Singhal et al., 2025; Nisonoff et al., 2024), reward fine-tuning (Rector-Brooks et al., 2024), and standard diffusion models (more diffusion/euler-maruyama steps, more capacity, longer training) is missing.
> >
>
> We thank the reviewer for the suggestion. We compare the proposed method with the inference alignment baselines (Singhal et al., 2025; Nisonoff et al., 2024) on the protein generation task (**with results included in Table 2**).  We also highlight that, for Boltzmann sampling, Table 1 compares training a standard diffusion model at the target temperature, to training a model at a higher temperature and then annealing to the target with our method. In both domains, our method compares favorably, achieving higher reward (for proteins), or sampling more accurately from the target (for the Ising model) relative to the baselines. Furthermore, we added a discussion of the cost and benefit tradeoffs of these methods **to section 4.3**, as well as **to the related works** (the reward fine-tuning section).
>
> We include a discussion of the trade-offs between various alignment methods here in the context of reward-tilting (these methods are not applicable directly to the annealing or product use-cases.):
>
> Our approach is training-free, compared to the expensive training phase required for reward fine-tuning. During inference, our method incurs computational cost through two main sources (when applied to reward fine-tuning): (i) it does inference with $M$ samples in parallel, and (ii) it requires $O(V)$ reward evaluations during each inference step (with vocabulary size $V$).
>
> - Reward fine-tuning methods: don’t incur either inference-time cost, and instead concentrate compute on the training phase. This means they largely have separate requirements for compute resources (such as memory) compared to our method. We emphasize that our inference time approach is not orthogonal to reward fine-tuning, and can be applied to a model which has undergone such fine-tuning.
> - FK Steering (Singhal et al., 2025) [7]: (with the base model as the proposal) incurs the $M$ parallel sample cost, but not the reward evaluation one
> - DG-Exact (Nisonoff et al., 2024) [8]: is an inference time method that incurs the reward-evaluation cost (ii) but does inference with a single sample (in fact, it is equivalent to our method with $M=1$ SMC sample). In practice our method has a run-time similar to DG-Exact, due to parallelizing over $M$.
>
> Therefore, our method can be seen as a training-free scheme which uses both strategies to improve the sample rewards, compared to the other inference-time baselines. This makes the method useful for tasks where additional inference compute can be spent to obtain higher quality samples
>
> [7] Raghav Singhal, Zachary Horvitz, Ryan Teehan, Mengye Ren, Zhou Yu, Kathleen McKeown, and
> Rajesh Ranganath. A general framework for inference-time scaling and steering of diffusion
> models. arXiv preprint arXiv:2501.06848, 2025.
>
> [8] Hunter Nisonoff, Junhao Xiong, Stephan Allenspach, and Jennifer Listgarten. Unlocking guidance
> for discrete state-space diffusion and flow models. arXiv preprint arXiv:2406.01572, 2024.
>
> > Limited novelty: FKC is adapted from diffusion to jump processes, the essence of FKC was already presented in Skreta et al. (2025). The re-weighting approach is immediate when transitioning from diffusion to jump processes. From this CTMC formulation the presented proofs are straight-forward derivations and rather mechanical.
> >
>
> Although our work builds on Feynman-Kac Correctors proposed in Skreta et al. (2025) [9], we respectfully disagree that its adaptation to the discrete case merits limited novelty. Our work presents the Feynman-Kac formula for jump processes in a readily available form for machine learning practitioners and incorporates it into the framework of discrete diffusion, providing concrete examples. We argue that the derivations require expertise in both Sequential Monte Carlo and discrete diffusion, so that they constitute a useful contribution for the broader community of readers.
>
> [9] Marta Skreta, Tara Akhound-Sadegh, Viktor Ohanesian, Roberto Bondesan, Alán Aspuru-Guzik,
> Arnaud Doucet, Rob Brekelmans, Alexander Tong, and Kirill Neklyudov. Feynman-kac correctors
> in diffusion: Annealing, guidance, and product of experts. arXiv preprint arXiv:2503.02819, 2025.

---

> ### Author Response · Authors · 2025-11-24
> **Author Response to Reviewer yXoH (part 4)**
>
> > What are the details on datasplits and hyperparameter selection procedures for all compared methods?
> >
>
> Thank you for this question, we have added the details of all the experiments in the updated manuscript and summarize these details below.
>
> - **For the Boltzmann distribution of the Ising model**, we parameterize the scores using the UNet architecture. For the hyperparameter search over the architecture settings, we sweep across different numbers of “base channels” and the sizes of time embeddings. For the optimizer we always use “AdamW” and search for the best learning rate and weight decay parameters. Additionally, we test two different ways to incorporate the time-dependency of the rate matrix: sinusoidal and linear. We optimize all the hyperparameters based on Wasserstein-2 distance between generated samples and the ground true samples (produced by the Swendsen-Wang algorithm). For the hyperparameters of DFKC, we optimize the number of integration steps and the number of particles. **We have added a description of all the hyperparameters to Appendix D.5 of the revised manuscript.**
> - **Amortized learning and Multi-constraint story generation**: Here, the main hyperparameter is the number of particles we use in DFKC, which was optimized based on the performance (MSE error, or PPL) on a set of prompts. Different seeds create different prompts. For amortized learning, this meant that for each run, different data for linear regression was fed into the prompt, while in the story task, the prompt contained a different set of constraints, randomly sampled from an existing bank of constraints. **Additional details are included in Appendix D.1.2 and D.3, for each experiment respectively.**
> - **Code generation task**: We optimize the number of particles, the inverse-temperature $\beta$, as well as the remasking strategy (random or low-confidence) used in DFKC. These were selected through grid search, on a small validation set (10 prompts from each dataset). The final evaluation results are reported on the remainder of the dataset (154 datapoints for HumanEval, and 417 points for MBPP). The sanitized datasplit for MBPP was used. **Additional details are included in appendix D.2 of the revised paper.**
> - **For the generation of protein sequences:** the main hyperparameter is the reward scale $\gamma$. This was selected by evaluating the average (unscaled) reward on a set of generated sequences (of smaller length - 10, and 20). We evaluated the method with 5 and 10 particles, **with results written in Table A5** in the Appendix. The best performing number of particles for each method was selected and reported in Table 2 of the revised paper. These details have been added to D.4.4 in the appendix.

---

### Official Review · Reviewer_ZDxR · 2025-10-31

**Soundness:** 3
**Presentation:** 3
**Contribution:** 1
**Rating:** 0
**Confidence:** 4

**Summary:**

The paper proposes **Discrete Feynman–Kac Correctors (DFKC)**, a general framework for **controlling discrete diffusion models at inference time** without retraining. It extends the continuous Feynman–Kac Correctors, previously applied to SDE-based diffusion models, to discrete-state continuous-time Markov chains (CTMCs) that underlie masked diffusion models. DFKC introduces a principled way to sample from modified distributions such as (i) temperature-annealed, (ii) product or geometric averages of multiple diffusion processes, and (iii) reward-tilted distributions incorporating external objectives. The approach leverages Sequential Monte Carlo (SMC) sampling to reweight and resample trajectories using quantities that can be computed directly from trained discrete diffusion models, requiring no fine-tuning or retraining.

Theoretically, the paper derives discrete analogues of the Feynman–Kac formula and forward Kolmogorov equations to justify these transformations. Empirically, DFKC is demonstrated on three domains: Ising model sampling, language modeling (amortized inference and multi-constraint text generation), and protein sequence design guided by reward functions. Across all tasks, DFKC improves controllability and sample quality compared to baselines like discrete diffusion models and existing guidance schemes. The work positions DFKC as a unifying inference-time control framework for discrete generative models, bridging probabilistic control theory and discrete diffusion processes.

**Strengths:**

1. **Clear and ambitious motivation:**
The paper tackles an important and current challenge in discrete generative modeling — how to control discrete diffusion models at inference time — an area that has received much less attention than its continuous counterpart.
2. **Elegant theoretical formulation:**
The authors successfully extend the Feynman–Kac Corrector framework from continuous stochastic differential equations to discrete-state continuous-time Markov chains (CTMCs), providing a clean mathematical generalization rooted in the Forward Kolmogorov Equation (FKE).
3. **No retraining required:**
A key practical advantage is that DFKC enables fine-grained control at inference time without any model retraining or fine-tuning, which is computationally attractive and broadly applicable.
4. **Algorithmic clarity and simplicity:**
The connection between Feynman–Kac theory and Sequential Monte Carlo (SMC) is presented clearly, leading to an implementable inference algorithm (Algorithm 1) that integrates reweighting and resampling seamlessly.
5. **Diverse and well-aligned experiments:**
The experiments are well-chosen to illustrate each theoretical component — annealing (Ising model), product-of-marginals (language model with multiple prompts), and reward-tilting (protein generation). The applications are thoughtfully aligned with the theoretical constructs.

**Weaknesses:**

1. **Lack of formal convergence guarantees:**
The paper derives correct and interpretable rate equations, but the convergence properties of the Sequential Monte Carlo (SMC) estimators in the discrete setting are not analyzed in depth. There are no explicit results on variance, bias, or sample complexity, which weakens the theoretical completeness of the
2. **Assumption-heavy derivations:**
Several key results rely on strong idealizations, such as perfect knowledge of the marginal ratios or ergodic and reversible Markov chains. In practice, these assumptions are difficult to satisfy in large discrete models like language or protein diffusion.
3. **Weak ablation and sensitivity analysis:**
The experiments do not analyze the impact of key hyperparameters, such as the number of SMC particles, temperature schedules, or the shape of the reward function. Such ablations would clarify robustness and sensitivity of the approach.
4. **Connection to related theory could be deepened:**
The paper does not sufficiently discuss recent stochastic control–based approaches that share conceptual similarities, such as the discrete stochastic control formulations in Pham et al. (2025) [1] or reinforcement-style discrete guidance models. Drawing clearer distinctions or theoretical parallels would strengthen the framing.
5. **Interpretation and intuition:**
While the mathematics is rigorous, the exposition is sometimes technically dense and abstract. The paper could benefit from more intuitive explanations or illustrative visualizations to make the discrete Feynman–Kac concept more accessible to a wider audience.

[1] Pham, L.T.N., et al. _“Discrete Markov Probabilistic Models: An Improved Discrete Score-Based Framework with Sharp Convergence Bounds under Minimal Assumptions.”_ Forty-second International Conference on Machine Learning (ICML, 2025).

**Questions:**

- Can the authors provide any formal convergence or variance bounds for the SMC estimator in the discrete case?
- Is the convergence to the target distribution guaranteed under approximate marginal ratios, or does the method risk degeneracy in high-dimensional state spaces?
- Can the authors clarify whether DFKC can be interpreted as a discrete control problem where the weighting term acts as a control cost?
- Could DFKC be applied to hybrid continuous–discrete models (e.g., molecular graphs or structured data)?
- Is there a potential to integrate DFKC with learning-based control, where the corrector parameters are adapted during training?
- Some derivations (e.g., Theorems 3.3–3.5) are technically dense. Could the authors provide a high-level algorithmic summary or schematic showing how the discrete Feynman–Kac updates interact with the diffusion process?
- Could a brief comparison table between continuous FKC and Discrete FKC formulations help readers understand the correspondence?

**Details Of Ethics Concerns:**

A significant concern arises regarding **potential duplication of content** between this submission (_Discrete Feynman–Kac Correctors_, under review for ICLR 2026) and a previously published workshop paper titled _“Discrete Feynman–Kac Correctors”_ by **Mohsin Hasan, Marta Skreta, Alan Aspuru-Guzik, Yoshua Bengio, and Kirill Neklyudov**, which appeared at the **ICML 2025 Workshop on AI4MATH**.

Both papers share the **same title** and display **substantial textual and mathematical overlap**:
- **Abstract and introduction:** nearly identical phrasing and structure, both presenting a framework for controlling discrete masked diffusion models via Sequential Monte Carlo (SMC) methods, inspired by the continuous Feynman–Kac Correctors.
- **Core theoretical results:** identical statements and numbering of Theorems 3.1 and 3.3, covering temperature annealing and product of marginals in discrete diffusion.
- **Equations and derivations:** equations (1)–(111) match exactly, including the Forward Kolmogorov equation, reverse-time rate matrix derivations, and appended proofs (Theorems B.1–B.3).
- **Experimental results:** both evaluate the same applications (Ising model annealing, amortized regression using `LLaDA`, and reward-guided protein design) and reproduce identical figures showing mean squared error versus dataset size and number of SMC samples.

Given this strong content match, the submission appears to be an **extended version of the ICML 2025 workshop paper**. While ICML workshops are non-archival and it is acceptable to build upon them for a full conference paper, proper disclosure is required under the ICLR submission policy. The ICLR submission currently does not acknowledge the existence of this prior version, which raises issues of transparency and self-plagiarism.

The authors should have explicitly disclosed the earlier ICML workshop publication in the “Ethics and Reproducibility” section of their ICLR submission. They should also have clarified the novelty of the present version—e.g., whether it introduces new theoretical results, experiments, or extended analysis beyond the workshop version.

If the overlap is purely textual and no substantial new contribution is present, this could constitute **dual submission or redundant publication**, which would be contrary to ICLR ethical guidelines.

---

> ### Comment · Reviewer_wKyB · 2025-11-13
> **Reviewer to Reviewer comment**
>
> Submitting an extended version of a non-archival workshop paper to a conference is standard practice at the top ML conferences.
> As a another reviewer of this paper, I think the evaluation should focus on the actual novelty and scientific quality of the submission.

---

> ### Author Response · Authors · 2025-11-24
> **Author Response to Reviewer ZDxR (part 1)**
>
> We thank the reviewer for their thorough review and detailed feedback. We are glad that the reviewer found our method to be theoretically elegant, and “computationally attractive”, while finding our experiments “well chosen”**.** We address the points raised in the review below.
>
>
> > **Lack of formal convergence guarantees:** The paper derives correct and interpretable rate equations, but the convergence properties of the Sequential Monte Carlo (SMC) estimators in the discrete setting are not analyzed in depth. There are no explicit results on variance, bias, or sample complexity, which weakens the theoretical completeness
> >
>
> We thank the reviewer for acknowledging the correctness and interpretability of the proposed results! Indeed, a deeper theoretical analysis of Feynman-Kac correctors is of great interest. For now, we provide the proof of the Feynman-Kac formula (**Theorem 2.1 of the revised paper**), which establishes the convergence of the time-discretization scheme. Establishing the convergence *bounds* on the sampling scheme requires development of more advanced tools akin to those proposed in [1,2]. These, however, would require an extensive theoretical investigation meriting an independent study and publication. **We have added a discussion on the theoretical guarantees to the related works section**.
>
> [1] Ren, Yinuo, Haoxuan Chen, Grant M. Rotskoff, and Lexing Ying. "How discrete and continuous diffusion meet: Comprehensive analysis of discrete diffusion models via a stochastic integral framework." *arXiv preprint arXiv:2410.03601* (2024).
>
> [2] Benton, Joe, Valentin De Bortoli, Arnaud Doucet, and George Deligiannidis. "Nearly $ d $-linear convergence bounds for diffusion models via stochastic localization." *arXiv preprint arXiv:2308.03686* (2023).
>
>
> > **Assumption-heavy derivations:** Several key results rely on strong idealizations, such as perfect knowledge of the marginal ratios or ergodic and reversible Markov chains. In practice, these assumptions are difficult to satisfy in large discrete models like language or protein diffusion.
> >
>
> There seems to be a potential misunderstanding, but our framework does not rely on any ergodicity or reversibility of Markov Chains. The main tool establishing the convergence of the scheme is the Feynman-Kac formula (formulated and proved in Theorem 2.1). Although, we indeed rely on perfectly learned marginal ratios, this is a common assumption for any diffusion model as it is the main requirement for reversing the noising process. We additionally emphasize that the empirical evaluations reinforce the validity of the method for trained large discrete diffusion models (such as LLaDA-8B-Instruct, and DPLM-650M).
>
>
> > **Weak ablation and sensitivity analysis:** The experiments do not analyze the impact of key hyperparameters, such as the number of SMC particles, temperature schedules, or the shape of the reward function. Such ablations would clarify robustness and sensitivity of the approach.
> >
>
> We thank the reviewer for this suggestion. To illustrate the robustness of the proposed method, for various experiments, we include ablations over the number of particles and sequence length:
>
> - For the amortized learning experiment, we ablate over the number of SMC particles in **Figure A1, in Appendix D.1**. We observe that there is a significant improvement when resampling (using >1 particles), and then a gradual decrease in performance as the number of particles increases past $8$ samples. One possible reason for this is that the parameter accuracy is only approximately correlated with the model likelihood, due to imperfect calibration in the pretrained model, which would mean that more accurately sampling from the product distribution does not improve parameter accuracy. However, we observe that any number of samples outperforms the joint prompting strategy.
> - For the protein experiments, we include results for 5 and 10 SMC particles **in Table A5 of  Appendix D.4.5.** We observe that at both 5 or 10 particles, our approach achieves a higher reward compared to the baselines.
> - In addition, for protein experiments, the plot of **Figure A4.a) of Appendix D.4.5** compares our method’s results across varying protein lengths $\{10, 50, 100\}$ as well as across a varying number of SMC particles $\{1, 5, 10\}$ for the ESM2 likelihood reward.

---

> ### Author Response · Authors · 2025-11-24
> **Author Response to Review ZDxR (part 2)**
>
> > **Connection to related theory could be deepened:** The paper does not sufficiently discuss recent stochastic control–based approaches that share conceptual similarities, such as the discrete stochastic control formulations in Pham et al. (2025) [1] or reinforcement-style discrete guidance models. Drawing clearer distinctions or theoretical parallels would strengthen the framing.
> >
>
> We thank the reviewer for this suggestion, we have added the corresponding connections to the related work section! We note that our method is an inference time strategy which can be applied on top of any method which produces a parameterized rate matrix, whether it be trained with RL control objectives, or more standard diffusion training.
>
> > **Interpretation and intuition:** While the mathematics is rigorous, the exposition is sometimes technically dense and abstract. The paper could benefit from more intuitive explanations or illustrative visualizations to make the discrete Feynman–Kac concept more accessible to a wider audience.
> >
>
> We thank the reviewer for this concrete suggestion on improving the presentation of the method! We have **included a graphical representation of the method in our updated Figure 1**; we hope this conveys our method more intuitively.
>
> ***
>
> **Questions:**
>
> We answer the reviewer’s questions below.
>
> > Can the authors provide any formal convergence or variance bounds for the SMC estimator in the discrete case?
> >
>
> and
>
> > Is the convergence to the target distribution guaranteed under approximate marginal ratios, or does the method risk degeneracy in high-dimensional state spaces?
> >
>
> We point the reviewer to **Theorem 2.1**, which provides a result for the soundness of our simulation scheme. As described earlier, more detailed results on convergence bounds, as well as results under imperfectly estimated score functions, constitute an important direction for future work.
>
> > Could DFKC be applied to hybrid continuous–discrete models (e.g., molecular graphs or structured data)?
> >
>
> Thank you for the interesting question. The DFKC algorithm can be applied to such models by computing weights based on the rate matrix component, while another method, such as (continuous) FKC [3] can be applied to the continuous score function. However, the theoretical properties of such a simulation scheme are not clear. Namely, whether this scheme samples from the correct marginals over the joint continuous-discrete space would need to be analyzed, and is a fruitful direction for future work.
>
> [3] Marta Skreta, Tara Akhound-Sadegh, Viktor Ohanesian, Roberto Bondesan, Alán Aspuru-Guzik,
> Arnaud Doucet, Rob Brekelmans, Alexander Tong, and Kirill Neklyudov. Feynman-kac correctors
> in diffusion: Annealing, guidance, and product of experts. arXiv preprint arXiv:2503.02819, 2025.
>
> > Can the authors clarify whether DFKC can be interpreted as a discrete control problem where the weighting term acts as a control cost?
> >
>
> and
>
> > Is there a potential to integrate DFKC with learning-based control, where the corrector parameters are adapted during training?
> >
>
> This is an intriguing question. It is unclear whether the DFKC algorithm can directly be interpreted as a control algorithm. However, being an SMC procedure, it can be combined with methods for training SMC proposals, as in [4, 5], which can be framed as solving an optimal control task. Investigating the nature of these control tasks, and how they relate to various RL fine-tuning procedures, would merit further investigation.
>
> [4] Jeremy Heng, Adrian N. Bishop, George Deligiannidis, and Arnaud Doucet. Controlled sequential
> monte carlo. The Annals of Statistics, 48(5), October 2020. ISSN 0090-5364. doi: 10.1214/
> 19-aos1914.
>
> [5] Stephen Zhao, Rob Brekelmans, Alireza Makhzani, and Roger Grosse. Probabilistic inference in
> language models via twisted sequential monte carlo, 2024.
>
> > Some derivations (e.g., Theorems 3.3–3.5) are technically dense. Could the authors provide a high-level algorithmic summary or schematic showing how the discrete Feynman–Kac updates interact with the diffusion process?
> >
>
> As mentioned before, we have **included a graphical representation of the method in our updated Figure 1**.
>
> > Could a brief comparison table between continuous FKC and Discrete FKC formulations help readers understand the correspondence?
> >
>
> We agree with the reviewer that a comparison table between DFKC and continuous FKC would be useful to understand their connection and **have included this in Appendix A.1**.
>
> ***

---

### Author Response · Authors · 2025-11-24
**Global Response to Reviews**

We thank all the reviewers for their feedback and time spent reviewing our submission. We were pleased that the method was described as “computationally attractive” and “broadly applicable” (reviewer ZDxR), and that the theoretical derivations were judged “rigorous and well-founded” (reviewer yXoH).

Following reviewer suggestions, we have significantly strengthened the manuscript: updating its presentation, the theoretical component, and the empirical study. In particular, we have updated the following key points:

- **Temperature annealing improves the performance on the code generation task (see section 4.2, and Figure 4).**  Based on comments from reviewers, we added a more challenging language modelling task. In particular we demonstrate that annealing with our method improves the accuracy in solving programming problems in the HumanEval and MBPP datasets, outperforming alternative sampling strategies.
- **Reward-tilted guidance allows for efficient optimization for thermostability of the protein sequences (see section 4.3, and Table 2).** To demonstrate the flexibility of our method, we have added experiments for protein generation using another reward function that measures thermostability (a desirable property of natural proteins). As with the likelihood experiments, our method generates sequences with a higher reward compared to the baselines.
- **Temperature annealing allows for sampling from the Ising model at the critical temperature and beyond (see Figure 2).** We have extended the Ising model experiments to the more challenging setting of sampling beyond the critical temperature, given initial samples in a high-temperature setting. The results demonstrate that our method outperforms the baselines past the critical temperature.
- **For protein generation and the Ising model, we have added requested metrics and comparisons to baselines.** As suggested by reviewer yXoH we have added new metrics in the evaluation of our protein generation task, mainly those measuring diversity, structural confidence, and novelty of sequences. These are included in Table 2, evaluated on  our method. This table also includes the baselines DG-Exact [1] and FK Steering [2]. Furthermore, for the Ising model task, we have added a diffusion sampler baseline in LEAPS [3] (displayed in Figure 2).
- **We proved convergence guarantees for discrete space continuous-time Markov chains (see Theorem 2.1 and its proof), and discussed the relevant literature on convergence bounds (see new paragraph in the Related Work section).** As requested by the reviewers, we prove that the Feynman-Kac formula holds for discrete states and continuous time. However, establishing the convergence *bounds* for the proposed method requires a separate extensive theoretical study, as we discuss in the revised related works section.

Significant additions to the text are highlighted in the revised submission.

We provide individual responses to each reviewer below. We hope that our responses address the major concerns of the reviewers and we remain open for more questions throughout the discussion period.

[1] Hunter Nisonoff, Junhao Xiong, Stephan Allenspach, and Jennifer Listgarten. Unlocking guidance
for discrete state-space diffusion and flow models. arXiv preprint arXiv:2406.01572, 2024.

[2] Singhal, Raghav, Zachary Horvitz, Ryan Teehan, Mengye Ren, Zhou Yu, Kathleen McKeown, and Rajesh Ranganath. "A general framework for inference-time scaling and steering of diffusion models." *arXiv preprint arXiv:2501.06848* (2025).

[3] Peter Holderrieth, Michael Samuel Albergo, and Tommi Jaakkola. LEAPS: A discrete neural sampler
via locally equivariant networks. International Conference on Machine Learning (ICML), 2025.
URL https://openreview.net/forum?id=Hq2RniQAET.

---

### Meta-Review · Area_Chair_pBM6 · 2026-01-06

**Summary:**

This paper is highly borderline. Reviewers recognized the work as technically sound and appreciated the clean extension of Feynman–Kac–based inference-time control to discrete diffusion models, as well as the breadth of empirical applications. The rebuttal substantially improved the manuscript by adding stronger experiments, clearer presentation, additional baselines, and more detailed evaluations, which resolved several concerns regarding empirical depth and clarity.

However, multiple reviewers continued to raise concerns about the limited methodological novelty relative to prior Feynman–Kac Corrector work and the absence of formal convergence, variance, or sample-complexity guarantees for the SMC estimator in the discrete setting. While these limitations are acknowledged by the authors as future work, they remain central to assessing the contribution at this stage. As a result, despite being close to the acceptance threshold, I recommend a weak reject, while strongly encouraging the authors to further brush up the paper by sharpening the novelty claims and strengthening the theoretical guarantees in a future revision.

**Reviewer Concerns:**

- Reviewer ZDxR:
Concerns about limited ablations, presentation density, and connections to related stochastic control and guidance literature were largely addressed by added experiments, figures, and expanded related-work discussion. However, the lack of formal convergence or variance guarantees for the SMC estimator in the discrete setting remains outstanding and is acknowledged as future work.

- Reviewer yXoH:
Requests for stronger empirical evidence, additional metrics, and more realistic applications were addressed through expanded Ising experiments, added code generation tasks, richer protein evaluation metrics, and new baselines. Nevertheless, concerns regarding limited novelty relative to prior Feynman–Kac Corrector work and the overall cost–benefit tradeoff compared to alternative alignment methods remain partially unresolved.

- Reviewer wKyB:
Concerns about missing comparisons, notation clarity, and the absence of critical-temperature Ising evaluations were fully addressed by added baselines, clarified notation, and extended experiments beyond the critical regime. No major outstanding concerns remain from this reviewer.

- Reviewer AQs3:
Questions about methodological scope and time-discretization interpretation were clarified, and experimental breadth was improved in the revision. Still, the absence of explicit theoretical convergence guarantees and skepticism about algorithmic novelty remain outstanding concerns.

**Reviewer Scores:**

- Reviewer ZDxR: After the original concern was resolved, the score was revised accordingly to score 6, the reviewer’s final assessment already reflects the discussion.

- Reviewer yXoH: Although the rebuttal strengthened the empirical results and clarified motivation, the reviewer’s core concerns regarding novelty and real-world impact would likely keep the score unchanged.

- Reviewer wKyB: This reviewer explicitly stated that the rebuttal addressed most questions satisfactorily and confirmed their original positive evaluation, so the score would remain unchanged.

- Reviewer AQs3: While acknowledging the clarifications and accepting that convergence guarantees are out of scope, the reviewer’s reservations about methodological novelty suggest the score would remain unchanged.

---

### Decision · Program_Chairs · 2026-01-26

Reject